# BOOSTING FOR PREDICTIVE SUFFICIENCY

**Abbavaram Gowtham Reddy**[†]**, Rajeev Verma**[‡]**, Celia Rubio-Madrigal**[†]**,**
**Krikamol Muandet**[†]***, Rebekka Burkholz**[†*]
[†]CISPA Helmholtz Center for Information Security      [‡]University of Amsterdam

## ABSTRACT

Out-of-distribution (OOD) generalization is a defining hallmark of truly robust and reliable machine learning systems. Recently, it has been empirically observed that existing OOD generalization methods often underperform on real-world tabular data, where hidden confounding shifts drive distribution shifts that boosting models handle more effectively. Earlier work attributes a part of boosting's success to variance reduction, handling missing covariates, feature selection, and connections to multicalibration. Complementary to these explanations, we uncover a crucial reason behind boosting's success in OOD generalization: its ability to identify environments created by hidden confounding shifts and maximize predictive performance within those environments. To this end, this paper introduces an information-theoretic notion called $\alpha$-predictive sufficiency and formalizes its connection to OOD generalization under hidden confounding shift. We show that boosting implicitly identifies suitable environments and produces an $\alpha$-predictive sufficient predictor. We validate our theoretical results through synthetic and real-world experiments and show that boosting achieves robust performance by identifying these environments and maximizing the mutual information between predictions and true outcomes.

## 1 INTRODUCTION

Generalization beyond the training distribution is central to trustworthy machine learning. Numerous methods have been proposed to enhance out-of-distribution (OOD) performance on data that differs from the in-distribution (ID) training data (Muandet et al., 2013; Arjovsky et al., 2019; Sagawa et al., 2019; Liu et al., 2021c; Zhou et al., 2022; Singh et al., 2024; Yang et al., 2024). These methods typically rely on assumptions such as invariance to ensure generalization beyond training environments. In practice, however, factors such as differing data-generating processes, selection bias, measurement error, and shifts in unobserved confounding variables often undermine the validity of these assumptions (Fan et al., 2014; Alabdulmohsin et al., 2023; Tsai et al., 2024; Liu & Cui, 2025; Prashant et al., 2025). Consequently, sophisticated methods for OOD generalization often underperform more traditional methods such as boosting, mixture-of-experts, and multi-layer perceptrons (Gulrajani & Lopez-Paz, 2021; Vedantam et al., 2021; Rosenfeld et al., 2022; Gardner et al., 2023; Liu et al., 2023; Nastl & Hardt, 2024). It is therefore crucial to understand the underlying mechanisms that enable traditional methods to generalize better under real-world distribution shifts (Fan et al., 2014; Liu & Cui, 2025; Reddy et al., 2026).

The nature of underlying distribution shifts guides the development of generalizable methods. Traditionally, it is assumed that distribution shifts are due to either *label shift* or *covariate shift*. Recent studies however reveal that *hidden confounding shift* (confounding variable is a variable that causally influences both label and covariates) is also prevalent in real-world data (see § 3). Based on these assumptions, existing approaches typically consider partitions of the data to capture the inherent heterogeneity of the underlying data distribution. Such partitioning—whether specified a priori or defined by experts—serves as the basis for different notions of generalization such as invariance (Arjovsky et al., 2019; Krueger et al., 2021; Creager et al., 2021), robustness (Sagawa et al., 2019), multicalibration (Kim et al., 2019; Wald et al., 2021; Gopalan et al., 2022; Wu et al., 2024a), and predictive information (Reddy et al., 2026).

---

[*]These authors share senior authorship.

A popular approach of defining such partitioning is to assign environment labels using metadata from the data-collection process. For instance, in housing price prediction, region identifiers such as zip codes or state/country names are commonly used as environment labels (Gardner et al., 2023). Similarly, in medical diagnostics, hospital IDs—reflecting differences in equipment, protocols, and patient populations—often serve as environment labels when training models to predict disease outcomes from lab-test data. Data categorization into different subpopulations or environments may lead to different performances (Liu et al., 2021a; Liu & Cui, 2025). When the underlying environment labels are not available or the readily available environment labels do not accurately represent the underlying data heterogeneity, recent methods focus on identifying *correct* subpopulations so that the invariance relationships between covariates and labels can be learned effectively (Liu et al., 2021a;b; Lin et al., 2022; Liu et al., 2024).

Due to its reliance on the data partitioning, OOD generalization problem is an algorithmic manifestation of the reference class problem (Hájek, 2007; Hu, 2025): *Given a single case (an individual, an event, a situation), which group or "reference class" should we use to assign the probability to that case?* For example, to predict the price of a house in New York City, one may consider the reference class to be the set of all houses in the New York City and attribute their average price as a prediction for the current house. Another possible reference class is the set of houses within a radius of ten kilometers. Hence, the prediction depends heavily on the partition we choose. If the model partitions the data "wrong", e.g., grouping patients by hospital ID rather than disease mechanism, we may get incorrect predictions for OOD data. This is particularly challenging when the distribution shifts are due to shifts in hidden confounders, as we discuss in § 3.

We hypothesize that ensemble methods, such as boosting, achieve strong OOD performance because they implicitly partition data into reference classes that align with hidden confounding shifts. Specifically, the clusters formed by leaf embeddings of boosted trees align with hidden confounding shifts, which contributes to improved robustness under distribution shifts induced by hidden confounding shifts. To formally investigate this, we define an information-theoretic notion called $\alpha$-*predictive sufficiency*. We first show the connection between $\alpha$-predictive sufficiency and OOD generalization under hidden confounding shifts. We then show that boosting can be viewed as an algorithm that learns an $\alpha$-predictive sufficient predictor. Since boosting returns $\alpha$-predictive sufficient predictors, boosting is effective in solving the OOD generalization problem under a hidden confounding shift. Unlike the traditional explanations for the success of boosting based methods, our explanations focus on the aspect of implicit identification of environments that lead to generalization under hidden confounding shift. As a special case, we extend our theoretical analysis to gradient boosting. Our contributions are summarized as follows.

- We define an information-theoretic notion of $\alpha$-predictive sufficiency. We then show its connection to OOD generalization under hidden confounding shift, expressed in terms of mutual information between ground truth labels and predictions (§ 4).

- We show that the boosting algorithm returns a predictor that is $\alpha$-predictive sufficient and, in doing so, boosting implicitly identifies environments corresponding to hidden confounding shifts (§ 5).

- Our experiments on synthetic and real-world data validate our claims that boosting implicitly captures hidden confounding shifts for generalization (§ 6).

## 2 RELATED WORK

**OOD Generalization Under Hidden Confounding Shift.** In recent years, out-of-distribution (OOD) generalization under hidden confounding has attracted considerable attention due to its prevalence in real-world data (Landeiro & Culotta, 2018; Alabdulmohsin et al., 2023; Tsai et al., 2024; Prashant et al., 2025). Solutions to this problem often include either adjusting for hidden confounder value (Alabdulmohsin et al., 2023; Tsai et al., 2024) or inferring hidden confounder value (Prashant et al., 2025) under proxy variable assumptions. Because the true confounder (parent of the outcome) is latent, achieving full invariance under hidden confounding is challenging. A practical alternative is to identify regions of the input distribution corresponding to hidden confounding shifts and deploy specialized predictors per region (Reddy et al., 2026).

When model architectures align with the underlying data-generating process, generalization may improve (Li et al., 2023; Wu et al., 2024b). Motivated by the conditional `if-then-else` structure of

visual attributes, Li et al. (2023) propose sparse mixture-of-experts (MoE) models for learning generalizable predictors. More generally, models of the form: $\mathbb{P}(Y \mid \mathbf{X}) = \sum_U \mathbb{P}(U \mid \mathbf{X})\mathbb{P}(Y \mid \mathbf{X}, U)$, where $U$ is a hidden confounder influencing both covariates $\mathbf{X}$ and label $Y$, can equivalently be interpreted as routing weights $\mathbb{P}(U \mid \mathbf{X})$ with per-region invariant predictions $\mathbb{P}(Y \mid \mathbf{X}, U)$. Consequently, several recent architectures for OOD robustness draw inspiration from MoE designs (Wu et al., 2024b; Prashant et al., 2025). In this work, we study how boosting provides structural inductive biases that enable generalization under hidden confounding shift. Although boosting is empirically effective in such settings, the mechanisms behind this success remain underexplored.

**Boosting and multicalibration.** Multicalibration, originally introduced for fairness (Hébert-Johnson et al., 2018), has recently been adapted for invariance learning (Wu et al., 2024a). Prior work establishes strong connections between multicalibration and boosting for regression (Globus-Harris et al., 2023; Wu et al., 2024a) and classification (Kim et al., 2019; Gopalan et al., 2022). Bayes-optimality guarantees for these methods rely on structural assumptions about the grouping functions used for multicalibration. For example, Wu et al. (2024a) show that multicalibration across source environments by grouping data using density-ratio functions between source and target joint distributions also implies multicalibration across target environments. These density ratios act as reference classes and isolate subpopulations most affected by shift, enabling invariance without explicit group annotations. However, this framework addresses covariate shifts in $\mathbb{P}(\mathbf{X})$ and concept shifts in $\mathbb{P}(Y \mid \mathbf{X})$, but does not explicitly accommodate shifts in $\mathbb{P}(\mathbf{X} \mid Y)$, which often arise under hidden confounding shifts. Similar approaches for inferring environments appear in (Liu et al., 2021a; Creager et al., 2021). While earlier work shows how level set boosting achieves invariance via multicalibration (Wu et al., 2024a), we show how boosting is linked to maximizing conditional informativeness as well as achieving invariance via predictive sufficiency.

**Reference Classes and Predictive Information.** Density-ratio groupings discussed above can be viewed as a surrogate for partitioning data by unobserved confounder values. While such groupings may not recover latent assignments exactly, they can highlight regions of equal confounder values. Recently, Reddy et al. (2026) showed that, OOD generalization can be framed as maximizing predictive information between true labels and predicted labels across environments that encode hidden confounding shifts. These environments should correspond to reference classes induced by the confounder values. We show that boosting maximizes predictive information within the reference classes induced by hidden confounding.

## 3 OOD GENERALIZATION UNDER HIDDEN CONFOUNDING SHIFT

In this section, we provide the necessary background on hidden confounding shifts and reference class to understand the rest of the paper. We use lowercase letters (e.g. $x$) for scalars, uppercase letters (e.g. $X$) for random variables, bold lowercase (e.g. $\mathbf{x}$) for scalar-valued vectors, bold uppercase (e.g. $\mathbf{X}$) for random vectors, and script letters (e.g. $\mathcal{X}$) for sets. Following previous work (Alabdulmohsin et al., 2023; Tsai et al., 2024; Prashant et al., 2025; Reddy et al., 2026), we model hidden confounding shift across environments using the

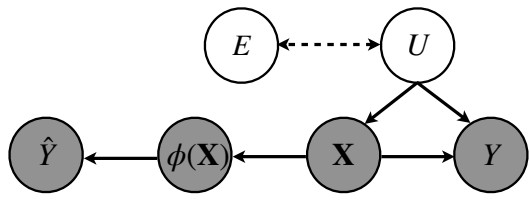

Figure 1: Causal graph for modeling hidden confounding shifts across environments.

causal graph $\mathcal{G}$ shown in Figure 1. $\mathcal{G}$ contains as nodes the covariate vector $\mathbf{X}$, the label $Y$, the environment variable $E$, a hidden confounder $U$, the feature extractor $\phi$, and the prediction $\hat{Y}$. Directed arrows among $\mathbf{X}, Y, U, E$ denote the direct causal influences in the true data-generating process. Directed arrows towards $\phi(\mathbf{X})$ and $\hat{Y}$ encode the structural (computational) dependencies that generate representations $\phi(\mathbf{X})$ from $\mathbf{X}$ and predictions $\hat{Y}$ from $\phi(\mathbf{X})$. A model's prediction $\hat{Y}$ can be obtained as $\hat{Y} = (f \circ \phi)(\mathbf{X})$ where $f$ is a function that makes predictions using the representations $\phi(\mathbf{X})$. Let $\mathcal{X}, \mathcal{Y}, \mathcal{U}, \mathcal{E}$ denote the domains of $\mathbf{X}, Y, U, E$ respectively. Let $E$ be a *discrete* variable that encodes distribution shifts of the *continuous* hidden confounder $U$ across environments. That is, for any two different environments $e, e' \in \mathcal{E}$, we have $\mathbb{P}^e(U) \neq \mathbb{P}^{e'}(U)$. We model $E$ as an unobserved variable because boosting methods do not require access to $E$ and work on the pooled data from all environments.

We denote the entropy of a random variable $X$ by $H(X) := -\mathbb{E}_X[\log(\mathbb{P}(X))]$ and the mutual information between two random variables $X, Y$ by $I(X;Y) := \mathbb{E}_{X,Y}[\log \frac{\mathbb{P}(X,Y)}{\mathbb{P}(X)\mathbb{P}(Y)}]$. Conditional entropy and conditional mutual information are defined similarly. For the theoretical analysis, we consider the mutual information $I(Y; \hat{Y})$ between the true label $Y$ and the predictive counterpart $\hat{Y}$ as the performance measure of a model. Following Federici et al. (2021) and Reddy et al. (2026), we measure the concept shift across environments with $I(Y; E \mid \phi(\mathbf{X}))$, which can also be viewed as a measure of invariance. Concretely, from the causal graph in Figure 1, we assume the causal mechanism $\mathbb{P}(Y \mid \mathbf{X}, U)$ is invariant across environments: $\mathbb{P}^e(Y \mid \mathbf{X}, U) = \mathbb{P}^{e'}(Y \mid \mathbf{X}, U) \ \forall e, e' \in \mathcal{E}$. If the representation $\phi(\mathbf{X})$ encodes the hidden confounder in the sense that there exists a measurable function $g$ with $U = g(\phi(\mathbf{X}))$ almost surely, then the causal mechanism of $Y$ given $\phi(\mathbf{X})$ is invariant as well: $\mathbb{P}^e(Y \mid \phi(\mathbf{X})) = \mathbb{P}^{e'}(Y \mid \phi(\mathbf{X}))$. Equivalently, the environment is independent of $Y$ given $\phi(\mathbf{X})$, so the conditional mutual information vanishes: $I(Y; E \mid \phi(\mathbf{X})) = 0$.

We now briefly review key definitions from causal graphical models (Pearl, 2009). A causal directed acyclic graph is a directed graph whose vertices correspond to random variables and directed arrows represent direct causal influences. A path in a graph is a sequence of distinct nodes connected by edges. A path is said to be directed if every edge aligns with the direction of the path. Along such a path, one may refer to parents, children, ancestors, and descendants in the usual sense. For instance, in the directed path: $X \to Y \to Z$, $X$ is the parent of $Y$, $Y$ is the child of $X$, $Z$ is the descendant of $X$, and $X$ is the ancestor of $Z$. Any three variables $X, Y, Z$ in a causal graph can form one of the three basic substructures: (i) a *chain* ($X \to Y \to Z$), (ii) a *fork* ($X \leftarrow Y \to Z$), and (iii) a *collider* ($X \to Y \leftarrow Z$). In both chains and forks, $X$ and $Z$ are marginally dependent but become independent conditional on $Y$. In a collider, $X$ and $Z$ are marginally independent, yet conditioning on the collider $Y$ (or its descendants) renders them dependent. A path $p$ is said to be *blocked* by a conditioning set $\mathcal{B}$ if either (a) $p$ contains a chain or fork with its middle node in $\mathcal{B}$, or (b) $p$ contains a collider such that neither the collider nor its descendants belongs to $\mathcal{B}$. Two nodes are conditionally independent given $\mathcal{B}$ only when every path between them is blocked by $\mathcal{B}$.

**Hidden Confounding Shift.** Distribution shifts are usually modeled through a shift in the distribution of a hidden variable $U$ and how $U$ influences other observed variables $\mathbf{X}, Y$. For instance, when $U \to Y$ and $U \not\to \mathbf{X}$, we observe lable shift (Tachet des Combes et al., 2020; Garg et al., 2020; Alexandari et al., 2020; Wu et al., 2021). When $U \not\to Y$ and $U \to \mathbf{X}$, we observe covariate shift (Gretton et al., 2009; Sugiyama & Kawanabe, 2012; Schneider et al., 2020). In this work, we consider the case where $U \to Y$ and $U \to \mathbf{X}$, which is more prevalent in real-world data and is referred to as a hidden confounding shift (Alabdulmohsin et al., 2023; Liu et al., 2023; Tsai et al., 2024; Prashant et al., 2025; Reddy et al., 2026).

**Reference classes.** As discussed in § 1 and § 2, the notion of reference class is crucial for OOD generalization if the generalization is achieved via multicalibration or predictive information. Here, we first formally state the definition of reference class, and environments induced by those reference classes (Definition 3.1), and then describe the crucial assumption of common confounder support.

**Definition 3.1** (Reference Classes and Environments). *Let $\mathcal{X} \subseteq \mathbb{R}^d$ be the covariate space and $\mathcal{S} \subseteq \{1, \ldots, d\}$ be an index set. For any $\mathbf{x} \in \mathcal{X}$, we write $\mathbf{x}_\mathcal{S}$ for the projection of $\mathbf{x}$ onto the coordinates in $\mathcal{S}$. The* reference class *of $\mathbf{x}$ with respect to $\mathcal{S}$, denoted by $[\mathbf{x}]_\mathcal{S}$, is defined as $[\mathbf{x}]_\mathcal{S} := \{\mathbf{x}' \in \mathcal{X} : \mathbf{x}'_\mathcal{S} = \mathbf{x}_\mathcal{S}\}$. On a finite dataset $\mathcal{D} = \{\mathbf{x}^{(1)}, \ldots, \mathbf{x}^{(n)}\}$, we define an equivalence relation $\mathbf{x} \sim_\mathcal{S} \mathbf{x}' \iff \mathbf{x}_\mathcal{S} = \mathbf{x}'_\mathcal{S}$, and the equivalence classes under the relation $\sim_\mathcal{S}$ are precisely the reference classes. In $\mathcal{D}$, the collection of all reference classes $\mathcal{E}_\mathcal{S} := \{[\mathbf{x}]_\mathcal{S} : \mathbf{x} \in \mathcal{D}\}$ forms a partition of $\mathcal{D}$, which we refer to as the* environment partition *of $\mathcal{D}$.*

**Assumption 3.1** (Common confounder support (Prashant et al., 2025)). *We assume that every confounder value that occurs in a test environment $e_{te}$ also occurs in the train environment $e_{tr}$ i.e., $\mathrm{supp}(\mathbb{P}(U|E = e_{te})) \subseteq \mathrm{supp}(\mathbb{P}(U|E = e_{tr}))$. In finite-sample terms, the set of confounder values possible in the test data is contained in the set of confounder values possible in the train data.*

Assumption 3.1 rules out target-only confounder values and ensures that the training data provide examples from all confounder regions that may be encountered at test time. Common confounder support is necessary for generalization because, under the standard invariance assumption that $\mathbb{P}(Y \mid \mathbf{X}, U)$ is shared across environments, we have the decomposition: $\mathbb{P}^{e_{te}}(Y \mid \mathbf{X}) = \sum_U \mathbb{P}^{e_{te}}(U \mid \mathbf{X})\mathbb{P}^{e_{tr}}(Y \mid U, \mathbf{X})$. From this decomposition, if $\mathbb{P}^{e_{te}}(U|\mathbf{X}) > 0$ and $\mathbb{P}^{e_{tr}}(U|\mathbf{X}) = 0$, then $\mathbb{P}^{e_{tr}}(Y \mid$

$U, \mathbf{X}$) cannot be identified from the training data for those values of $U$, leading to fundamental limits on generalization when the model encounters unseen confounding patterns at test time.

# 4  PREDICTIVE SUFFICIENCY FOR OOD GENERALIZATION

In this section, we use the notion of maximizing predictive information as the target metric for OOD generalization under hidden confounding shift. We then define the notion of $\alpha$-predictive sufficiency (Definition 4.3) as a proof concept that relates to this target metric, and using the decomposition results recently proposed by Reddy et al. (2026), we prove that achieving $\alpha$-predictive sufficiency is linked to achieving generalization under hidden confounding shift (Proposition 4.1). We will leverage these insights to explain the success of boosting methods for OOD generalization under the hidden confounding shift. We begin by formally defining predictive information.

**Definition 4.1** (Predictive Information). *The predictive information between true outputs $Y$ and model predictions $\hat{Y}$ is defined as the mutual information $I(Y; \hat{Y})$.*

In a recent work, Reddy et al. (2026) show that predictive information admits an information-theoretic decomposition into interpretable components, namely *variation* $I(\phi(\mathbf{X}); E \mid Y)$, *feature shift* $I(\phi(\mathbf{X}); E)$, *label shift* $I(Y; E)$, *concept shift* $I(Y; E \mid \phi(\mathbf{X}))$, *conditional informativeness* $I(\phi(\mathbf{X}); Y \mid E)$, and *residual* $I(\phi(\mathbf{X}); Y \mid \hat{Y})$. Moreover, under hidden confounding shift, this decomposition simplifies as follows:

$$I(Y; \hat{Y}) = I(Y; \phi(\mathbf{X}) \mid E) - I(Y; \phi(\mathbf{X}) \mid \hat{Y}) \tag{1}$$

Minimizing the residual term implies that all the information contained in $\phi(\mathbf{X})$ is utilized by the function $f$ so that no residual information is left in $\phi(\mathbf{X})$. Conditional informativeness motivates maximizing the mutual information between representations and labels within each environment corresponding to the reference class induced by hidden confounder values. Model architectures such as boosting trees and MoE are especially good at modeling such environment-specific requirements (Wu et al., 2024b; Prashant et al., 2025; Li et al., 2023). Our goal in this work is to explain this puzzling empirical phenomenon and provide a better theoretical understanding of the mechanisms behind the success of boosting methods.

**Definition 4.2** (Prediction Error). *We define the prediction error as the difference between the true label $Y$ and the model prediction $\hat{Y}$: $Y - \hat{Y}$.*

In regression tasks, the quantity $Y - \hat{Y}$ has a natural interpretation as the prediction error between two scalar values. In classification tasks, we treat $\hat{Y}$ as the predicted probability instead of the predicted label so that $Y - \hat{Y}$ corresponds to the probability residual. To understand how the prediction error manifests in boosting, consider the example of gradient boosting. In gradient boosting, weak learners are trained by fitting pseudo-residuals, defined as the negative gradient of the loss with respect to the model output. These pseudo-residuals coincide with the raw prediction error $Y - \hat{Y}$ in common settings such as regression with mean squared loss and binary classification with cross-entropy loss. In the latter case, when the model outputs logits, the resulting pseudo-residuals take the form $Y - \hat{Y}$, where $\hat{Y}$ denotes the predicted probability.

**Definition 4.3** ($\alpha$-Predictive Sufficiency). *For $\alpha \geq 0$, a prediction $\hat{Y}$ is $\alpha$-predictive sufficient for $Y$ across environments $E$ if $I(Y - \hat{Y}; E \mid \hat{Y}) \leq \alpha$.*

Intuitively, 0-predictive sufficiency implies that the prediction error $Y - \hat{Y}$ is independent of $E$ for each outcome value $\hat{Y}$. Since $E$ encodes the information about $U$ (a direct parent of $Y$), achieving predictive sufficiency implies that the predictor $\hat{Y}$ relies on $\mathbf{X}$ and $Y$ through model training to implicitly account for the impact of $U$. This helps in learning a predictor capable of OOD generalization. As a first technical result, we prove that $\alpha$-predictive sufficiency naturally relates to the predictive information in the following proposition.

**Proposition 4.1.** *For a covariate vector $\mathbf{X}$, label $Y$, with causal structure $\mathbf{X} \rightarrow Y$, an environment variable $E$, a feature extractor $\phi$, and prediction $\hat{Y}$, $\alpha$-predictive sufficiency of $\hat{Y}$ for $Y$ across environments $E$ can be decomposed as follows:*

$$I(Y - \hat{Y}; E \mid \hat{Y}) = -I(Y; \phi(\mathbf{X}) \mid E, \hat{Y}) + I(Y; \phi(\mathbf{X}) \mid \hat{Y}) + I(Y; E \mid \phi(\mathbf{X})). \tag{2}$$

Proofs are presented in Appendix § A. Based on the causal graph in Figure 1, $I(Y; \phi(\mathbf{X}) \mid E, \hat{Y})$ is a lower bound on the conditional informativeness $I(Y; \phi(\mathbf{X}) \mid E)$ (see Appendix A.1 for a formal proof). Intuitively, conditioning on $\hat{Y}$ does not increase mutual information between $Y$ and $\phi(\mathbf{X})$ because $\hat{Y}$ is neither a collider nor belongs to a set of descendants of any colliders in the path between $Y$ and $\phi(\mathbf{X})$. Residual term from Equation 1 appears as is in Equation 2, and in addition we have the concept shift term, whose minimization is crucial for achieving invariance (Liu et al., 2023; Reddy et al., 2026). From Equations 1 and 2, we note the crucial observation: *achieving $\alpha$-predictive sufficiency for a small $\alpha$ is closely linked to maximizing predictive information under hidden confounding shift.* Building on this, we now show that gradient boosting returns an $\alpha$-predictive sufficient predictor.

## 5 BOOSTING FOR PREDICTIVE SUFFICIENCY

In this section, we leverage the insights from the previous section to argue that boosting returns $\alpha$-predictive sufficient predictor. On a mechanistic level, boosting iteratively combines several weak learners to form a strong learner. Weak learners usually take the form $h : \mathbf{X} \to \mathbb{R}; h \in \mathcal{H}$ where $\mathcal{H}$ is a hypothesis space of weak learners. For example, gradient boosting starts with a base predictor $\hat{Y}_0(\mathbf{X}) = h_0(\mathbf{X})$ (commonly a constant), and at each iteration $t$, it finds a new weak learner $h_t(\mathbf{X})$ that best fits the current pseudo-residuals $R_t(\mathbf{X}, Y)$ (negative gradients of the loss with respect to the output) and adds $\eta h_t(\mathbf{X})$ to the previous predictor $\hat{Y}_{t-1}(\mathbf{X})$ to get the current predictor $\hat{Y}_t(\mathbf{X}) = \hat{Y}_{t-1} + \eta h_t(\mathbf{X})$, where $\eta$ is the learning rate. To avoid notational clutter, we use $R_t$ and $h_t$ to denote the random variables $R_t(\mathbf{X}, Y)$ and $h_t(\mathbf{X})$.

There are several notions of weak learning (Natekin & Knoll, 2013; Schapire & Freund, 2013; Mayr et al., 2014; Bentéjac et al., 2021; Globus-Harris et al., 2023; Wu et al., 2024a), but on an intuitive level, a weak learner is a predictor that performs slightly better than random guessing or a constant predictor. In the same spirit, we begin by defining an information-theoretic weak learner below.

**Definition 5.1** ($\gamma$-approximate weak learner). *For any environment $e \in \mathcal{E}$, we say that the hypothesis class $\mathcal{H}$ satisfies the $\gamma$-approximate weak learning condition in the sense that whenever the Bayes predictor $Y^*$, defined as $Y^*(\mathbf{X}) = \mathbb{P}(Y \mid \mathbf{X})$, yields strictly more predictive information than a baseline predictor $b_e$ by a margin $\gamma$, i.e.,*

$$I(Y; Y^* \mid \mathbf{X} \in e) \geq \kappa(b_e) + \gamma$$

*where $\kappa(b_e) := I(Y; b_e \mid \mathbf{X} \in e)$. Then, there exists an $h \in \mathcal{H}$ that yields strictly more predictive information than a baseline predictor $b_e$ by the same margin $\gamma$, i.e.,*

$$I(Y; h(\mathbf{X}) \mid \mathbf{X} \in e) \geq \kappa(b_e) + \gamma.$$

In Definition 5.1, $\kappa(b_e) = 0$ for a constant predictor $b_e$. In what follows, we consider the constant predictor $b_e$ as a baseline predictor. In the context of gradient boosting, at each iteration $t$, the weak learning condition takes the form: $I(Y; h_t) \geq \gamma$, because the entire data is used at each iteration instead of any environment-specific data. Even though gradient boosting fits pseudo residuals $R_t$ at each iteration, leading to $I(R_t; h_t) \geq \gamma$, under mild assumptions, we prove that $I(Y; h_t) \geq \gamma$ holds in gradient boosting (see Appendix B for formal proof). We next state some standard assumptions before stating our main result.

**Assumption 5.1** (Existence of reweighted distributions). *At each round $t$, the exists a $p \in (0, 1)$ such that the boosting algorithm chooses reweighted distributions $\mathcal{D}_t(v)$ (e.g., based on model outputs/level sets/residuals) such that $\mathbb{P}(\hat{Y}_t = v) \geq p$ for all $t$.*

Assumption 5.1 avoids degenerated reweighting at each step and ensures non-trivial information is gained at each round $t$ when combined with Assumption 5.2 as described below. In the following, we make a conditional weak-learning assumption to ensure that each weak learner $h_t$ contributes a nontrivial amount of *new predictive information* beyond what is already captured by the previous weak learners $h_0, \ldots, h_{t-1}$.

**Assumption 5.2** (Conditional $\gamma$-approximate weak learning). *We assume that there exists a constant $\gamma > 0$ such that for all iterations $t$, $I(Y; h_t \mid h_0, \ldots, h_{t-1}) \geq \gamma$.*

---

**Algorithm 1** Standard boosting algorithm

---

**Require:** Step size $\eta$, base predictor $\hat{Y}_0$, hypothesis class $\mathcal{H}$, reweighting rule to obtain $\mathcal{D}_t$
1: Initialize $\hat{Y}_0 \leftarrow h_0, t \leftarrow 0$             ▷ $h_0$ is often a constant predictor
2: **while** training error decreases **do**
3:      Find weak learner $h_{t+1} \in \mathcal{H}$ that achieves $I(Y; h_{t+1}(\mathbf{X})) \geq \gamma$ under distribution $\mathcal{D}_t$
4:      Update predictor: $\hat{Y}_{t+1} \leftarrow \hat{Y}_t + \eta h_{t+1}(\mathbf{X})$
5:      Update distribution $\mathcal{D}_{t+1}$ using the reweighting rule and increment $t$ by 1.
6: **end while**

---

**Assumption 5.3** (Strong learner is a deterministic function of weak learners). *At any round $t$, $\hat{Y}_t$ is a deterministic function of $(h_0, \ldots, h_t)$.*

Assumption 5.3 is required to decompose the predictive information $I(Y; \hat{Y})$ into contributions from each individual weak learner. In practice, this assumption holds as the models are usually deterministic once they are trained. For our analysis, we consider the standard boosting algorithm described in Algorithm 1. We now show that this boosting algorithm can return an $\alpha$-predictive sufficient predictor after training for a certain number of time steps.

**Theorem 5.1.** *Under Assumptions 5.1-5.3, there exists a finite $T < \infty$ such that the predictor $\hat{Y}_t$ learned by Algorithm 1 after $t \geq T$ rounds is $\alpha$-predictive sufficient. The lower bound is given by*

$$T = \frac{H(Y) - H(Y \mid \mathbf{X}, E) - \alpha - I(Y; \hat{Y}_0)}{p \cdot \gamma}.$$

*When the environment $E$ is unknown, the same result holds by setting $H(Y \mid \mathbf{X}, E) = 0$.*

In particular, Theorem 5.1 guarantees that the iterative boosting algorithm converges in finite number of iterations, and lead to the $\alpha$-sufficient predictor.

We now prove the correspondence between a boosting model's leaf embeddings and the hidden confounder variable $U$. Since leaf embeddings in boosting model correspond to both outcome and input representations, we show how boosting achieves $H(U \mid \hat{Y}_T) \leq \beta$ for some $\beta > 0$ such that after the time step $T$, the uncertainty in $U$ given the predictions $\hat{Y}_T$ is less than or equal to $\beta$. We start with the assumption below.

**Assumption 5.4** (Structural preservation of $U$-information). *Let $\hat{Y}_t$ denote the predictor produced by gradient boosting after $t$ iterations. We assume that there exists a constant $c > 0$ for all finite $t$ such that $I(U; \hat{Y}_t) \geq c \cdot I(Y; \hat{Y}_t)$*

Motivated by the causal graph in Figure 1, Assumption 5.4 requires that $\phi(\mathbf{X})$ and the learning procedure preserve the components of $\mathbf{X}$ that carry $U$-signal, and the predictor actually uses those components rather than discarding them.

**Corollary 5.1.** *Under the Assumptions 5.1-5.4, there exists a finite $T < \infty$ such that the predictor $\hat{Y}_t$ learned by Algorithm 1 after $t \geq T$ rounds satisfies $H(U \mid \hat{Y}_t) \leq \beta$ for a small $\beta$. The lower bound on $T$ is given by*

$$T = \frac{H(U) - \beta - c \cdot I(Y; \hat{Y}_0)}{c \cdot p \cdot \gamma}$$

In Appendix B, we show similar theoretical bounds for the gradient boosting algorithm.

## 6   EXPERIMENTAL RESULTS

We perform experiments on both synthetic and real-world datasets to empirically explain how boosting excels at OOD generalization under hidden confounding shifts. Specifically, we compare the performance of boosting methods in terms of performance, predictive information, and predictive sufficiency. Additional results on both synthetic and Tableshift (Gardner et al., 2023) benchmark datasets are presented in Appendix § C.

**Methods:** We experiment on two standard boosting algorithms: CatBoost (Dorogush et al., 2018) and XGBoost (Chen & Guestrin, 2016). We use t-SNE (Maaten & Hinton, 2008) and PCA (Abdi & Williams, 2010) as dimensionality reduction methods for visualizations. We perform hyperparameter tuning to choose the best model when comparing the performance of models.

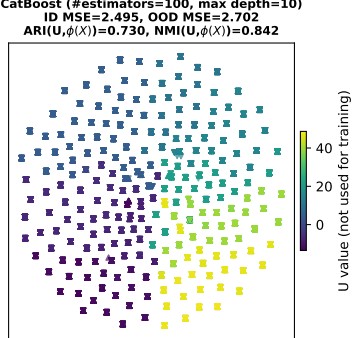

Figure 2: CatBoost representations with respect to hidden confounder value.

**Evaluation Metrics:** We consider the test accuracy to evaluate the performance of models in a classification setting, and test mean squared error (MSE) in a regression setting. We use the Adjusted Rand Index (ARI) and the Normalized Mutual Information (NMI) to evaluate the goodness of clustering (usually with respect to the hidden confounding variables in synthetic data). We evaluate predictive sufficiency and predictive information and compare the performance of models related to these measures. Mutual information is computed using the Kraskov-Stögbauer-Grassberger (KSG) estimator (Kraskov et al., 2004) from the nonparametric entropy estimation toolbox (Steeg & Galstyan, 2011; 2013).

**Synthetic Experiment 1:** We use linear structural equations to generate synthetic data following the causal graph: $U \to X, U \to Y, X \to Y, U \sim \mathcal{N}(\mu_e, \sigma_e)$ where $e \in \{1, 2, \ldots, 10\}$, $\mu_e$ is sampled randomly between $-50$ and $+50$, and $\sigma_e$ is set to $0.5$. Each environment has $500$ samples. We shift $\mu_e$ by a small fraction to induce a distribution shift where the test data lives. We train CatBoost and XGBoost on this dataset. To get the representations from these boosting methods, we obtain leaf embeddings for each data point as a vector of length $d$, where $d$ is the number of trees in the model. We observe that *models that achieve low MSE are those whose rep-*

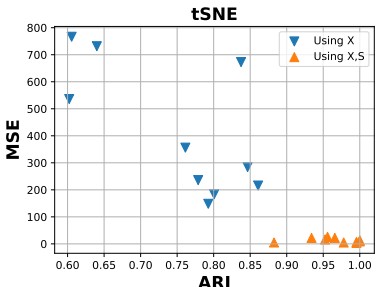

Figure 3: XGBoost ARI vs. MSE. Markers indicate results for different random seeds.

*resentations are aligned with hidden confounder values.* As shown in Figure 2, the clusters of representations of a trained CatBoost model align with the hidden confounder value. For similar visualizations for various combinations of dimensionality reduction techniques, values of the number of estimators (trees), and the maximum depth hyperparameters of CatBoost and XGBoost models, see the results in Figures C1-C4 in Appendix C.

**Synthetic Experiment 2:** Next, we consider a causal graph of $U_1 \to X, U_1 \to Y, X \to Y, U_2 \to S, U_2 \to Y, U_1 \sim \mathcal{N}(\mu_e, \sigma_e), U_2 \sim \mathcal{N}(\mu_f, \sigma_f)$. We generate 10 environments, each containing 50 samples. Note that $S$ does not causally influence the outcome $Y$. When XGBoost is only trained with $X$ as input, it fails to capture the underlying shifts in hidden confounder values because neither $U_2$ nor $S$ are observed during training; see Figure 3 where ARI is low and MSE is high. However, when XGBoost is trained using both $X$ and $S$ as inputs, we observe low MSE and high ARI. This result explains the observed phenomena that *adding additional covariates helps in generalization performance* as observed in (Nastl & Hardt, 2024). Any covariate that acts as a proxy for an unobserved confounder helps improve generalization performance.

**Real-world Data - California Housing Dataset:** In this dataset, the goal is to predict the *median house price* based on the features *median income, house age, rooms, bedrooms, population, occupancy, latitude, longitude*. We simulate an artificial hidden confounding shift using observed covariates and the outcome. Specifically, we use a combination of the values of *median income, house age, and median house price* to induce a distribution shift. Notably, we ensure that the values of hidden confounders at test time belong to the set of hidden confounder values at train time (Assumption 3.1). The results in Figure 4 clearly show clustering of PCA representations of XGBoost leaf embeddings with respect to hidden confounder values.

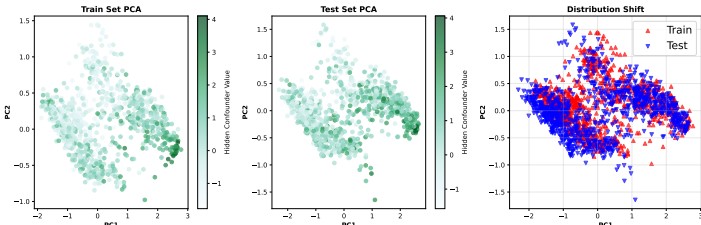

Figure 4: California housing dataset. XGBoost model representations are clustered according to hidden confounder values. There is a common confounder support between the train and test data.

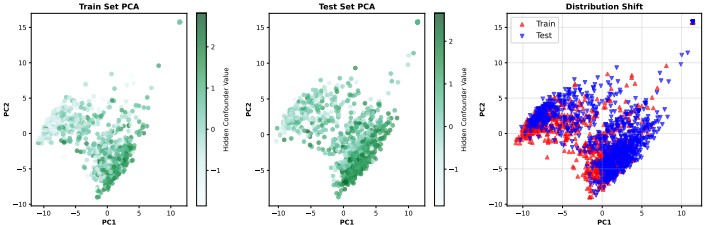

Figure 5: 20 Newsgroups dataset. XGBoost model representations are clustered according to the hidden confounder values. There is a common confounder support between the train and test data.

**Real-world Data - 20 Newsgroups:** The goal is to predict the news category of a document from its raw text. Documents are vectorized using TF-IDF and subsequently reduced with Truncated SVD to have 200 features. We simulate an artificial hidden confounding shift between train and test data using a combination of features such as document length, keyword indicator, and the outcome (class index). Similar to the previous experiment, we ensure that the common confounding support (Assumption 3.1) holds. Figure 5 shows a clear association between clustering of PCA representations of XGBoost leaf embeddings and the clusters associated with hidden confounder values.

Table 1: Comparison of XGBoost and CatBoost on real-world datasets.

| Method | California Housing | | | 20 Newsgroups | | |
| | MSE | Pred. Info. | Pred. Suffi. | Accuracy | Pred. Info. | Pred. Suffi. |
|---|---|---|---|---|---|---|
| XGBoost | $0.31 \pm 0.00$ | $0.47 \pm 0.03$ | $\mathbf{0.00 \pm 0.00}$ | $62.35 \pm 0.40$ | $0.27 \pm 0.10$ | $0.03 \pm 0.00$ |
| CatBoost | $\mathbf{0.29 \pm 0.00}$ | $\mathbf{0.56 \pm 0.10}$ | $\mathbf{0.00 \pm 0.00}$ | $\mathbf{62.61 \pm 0.00}$ | $\mathbf{0.68 \pm 0.08}$ | $\mathbf{0.00 \pm 0.00}$ |

**Comparison of Performance and Predictive Information:** Finally, we compare the performance of XGBoost and CatBoost and observe the underlying predictive information and predictive sufficiency values. It is evident from the results, shown in Table 1, that higher predictive information implies better performance, and a lower predictive sufficiency value means better performance. These results corroborate our theoretical claims.

## 7 LIMITATIONS, CONCLUSIONS, AND FUTURE WORK

We reframed OOD generalization under hidden confounding as a reference-class inference problem and introduced $\alpha$-predictive sufficiency as an information-theoretic target that characterizes when predictors transfer across environments. We prove that standard boosting algorithms return $\alpha$-predictive sufficient predictors in finitely many rounds, thereby implicitly inferring environments and maximizing predictive information, explaining their strong OOD behavior beyond variance reduction or feature selection alone. Empirically, across synthetic and real-world tabular tasks, boosting's learned representations cluster by hidden confounders and achieve high predictive information with low predictive-sufficiency residuals, aligning with the theory and yielding robust OOD performance. These results provide a principled account of why boosting often outperforms specialized OOD methods. We see this work as a foundation for new OOD algorithms that estimate or regularize $\alpha$, relax common-support assumptions, and extend predictive-sufficiency guarantees beyond tabular settings.

## ETHICS AND REPRODUCIBILITY STATEMENT

All authors have read and agree to adhere to the ICLR Code of Ethics. This work complies with all ethical guidelines outlined therein. Proofs of the theoretical results are presented in the appendix. The code and instructions to reproduce the results are provided in `https://github.com/gautam0707/Boosting_for_predictive_sufficiency`.

## ACKNOWLEDGMENTS

The authors gratefully acknowledge the Gauss Centre for Supercomputing e.V. for funding this project by providing computing time on the GCS Supercomputer JUWELS at Jülich Supercomputing Centre (JSC). We also gratefully acknowledge funding from the European Research Council (ERC) under the Horizon Europe Framework Programme (HORIZON) for proposal number 101116395 SPARSE-ML.

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

APPENDIX

## A  PROOFS OF THEORETICAL RESULTS

**Proposition 4.1.** *For a covariate vector* $\mathbf{X}$, *label* $Y$, *with causal structure* $\mathbf{X} \to Y$, *an environment variable* $E$, *a feature extractor* $\phi$, *and prediction* $\hat{Y}$, $\alpha$-*predictive sufficiency of* $\hat{Y}$ *for* $Y$ *across environments* $E$ *can be decomposed as follows:*

$$I(Y - \hat{Y}; E \mid \hat{Y}) = -I(Y; \phi(\mathbf{X}) \mid E, \hat{Y}) + I(Y; \phi(\mathbf{X}) \mid \hat{Y}) + I(Y; E \mid \phi(\mathbf{X})). \quad (2)$$

*Proof.* Given $\hat{Y}$, the mapping $Y \to Y - \hat{Y}$ is bijective. Hence $I(Y - \hat{Y}; E \mid \hat{Y}) = I(Y; E \mid \hat{Y})$. Hence, by expanding $I(Y; E \mid \hat{Y})$, we have the following.

$I(Y; E \mid \hat{Y}) = I(Y; E, \phi(\mathbf{X}) \mid \hat{Y}) - I(Y; \phi(\mathbf{X}) \mid E, \hat{Y})$
    (Introducing $\phi(\mathbf{X})$ using the chain rule of mutual information)
$= I(Y; \phi(\mathbf{X}) \mid \hat{Y}) + I(Y; E \mid \hat{Y}, \phi(\mathbf{X})) - I(Y; \phi(\mathbf{X}) \mid E, \hat{Y})$
    (Expanding the first term using the chain rule of mutual information)
$= I(Y; \phi(\mathbf{X}) \mid \hat{Y}) + I(Y, \hat{Y}; E \mid \phi(\mathbf{X})) - I(\hat{Y}; E \mid \phi(\mathbf{X})) - I(Y; \phi(\mathbf{X}) \mid E, \hat{Y})$
    (Introducing $\hat{Y}$ using the chain rule of mutual information)
$= I(Y; \phi(\mathbf{X}) \mid \hat{Y}) + I(Y, \hat{Y}; E \mid \phi(\mathbf{X})) - I(Y; \phi(\mathbf{X}) \mid E, \hat{Y})$
    ($I(\hat{Y}; E \mid \phi(\mathbf{X}))$ is zero because $\hat{Y} \perp\!\!\!\perp E \mid \phi(\mathbf{X})$ from the causal graph in Figure 1)
$= I(Y; \phi(\mathbf{X}) \mid \hat{Y}) + I(Y; E \mid \phi(\mathbf{X})) + I(\hat{Y}; E \mid \phi(\mathbf{X}), Y) - I(Y; \phi(\mathbf{X}) \mid E, \hat{Y})$
    (Expanding the middle term using the chain rule of mutual information)
$= I(Y; \phi(\mathbf{X}) \mid \hat{Y}) + I(Y; E \mid \phi(\mathbf{X})) - I(Y; \phi(\mathbf{X}) \mid E, \hat{Y})$
    ($I(\hat{Y}; E \mid \phi(\mathbf{X}), Y)$ is zero because $\hat{Y} \perp\!\!\!\perp E \mid \phi(\mathbf{X}), Y$ from the causal graph in Figure 1)

$\square$

**Theorem 5.1.** *Under Assumptions 5.1-5.3, there exists a finite* $T < \infty$ *such that the predictor* $\hat{Y}_t$ *learned by Algorithm 1 after* $t \geq T$ *rounds is* $\alpha$-*predictive sufficient. The lower bound is given by*

$$T = \frac{H(Y) - H(Y \mid \mathbf{X}, E) - \alpha - I(Y; \hat{Y}_0)}{p \cdot \gamma}.$$

*When the environment* $E$ *is unknown, the same result holds by setting* $H(Y \mid \mathbf{X}, E) = 0$.

*Proof.* From Assumption 5.3, $\hat{Y}_T$ is a deterministic function of $(h_0, h_1, h_2, \ldots, h_T)$, and $I(Y; \hat{Y}_0) = I(Y; h_0)$ because $\hat{Y}_0$ is initialized to $h_0$ before boosting training loop. Now we have the following:

$$I(Y; \hat{Y}_T) \leq I(Y; h_0, \ldots, h_T) = I(Y; \hat{Y}_0) + \sum_{t=1}^{T} I(Y; h_t \mid h_1, \ldots, h_{t-1}) \quad (3)$$

Consider any round $t \geq 1$, and let $S := (h_1, \ldots, h_{t-1})$. Then, we have

$$I(Y; h_t \mid S) = \mathbb{E}_s[I(Y; h_t \mid S = s)] = \mathbb{E}_s[I(Y; h_t \mid S = s)\mathbb{1}_{s \in S_t}] \geq p \cdot \gamma. \quad (4)$$

From Equations (3) and (4) we have the following.

$$I(Y; \hat{Y}_T) \leq I(Y; \hat{Y}_0) + T \cdot p \cdot \gamma \quad (5)$$

Given $\hat{Y}$, $Y - \hat{Y}$ and $Y$ have one-to-one correspondence. Hence, it follows that: $I(Y - \hat{Y}; E \mid \hat{Y}) = I(Y; E \mid \hat{Y})$. We next use the following simple upper bound on $I(Y; E \mid \hat{Y}_T)$:

$$I(Y; E \mid \hat{Y}_T) = H(Y \mid \hat{Y}_T) - H(Y \mid \hat{Y}_T, E) \leq H(Y \mid \hat{Y}_T), \quad (6)$$

and hence $I(Y; E \mid \hat{Y}_T) \leq H(Y \mid \hat{Y}_T)$. Furthermore, since $H(Y \mid \hat{Y}_T) = H(Y) - I(Y; \hat{Y}_T)$, we have: $I(Y; E \mid \hat{Y}_T) \leq H(Y) - I(Y; \hat{Y}_T)$ Now to get $I(Y; E \mid \hat{Y}_T) \leq \alpha$, it is sufficient to have $H(Y) - \alpha \leq I(Y; \hat{Y}_T)$, and following from Equation 5, it follows that

$$I(Y; \hat{Y}_0) + T \cdot p \cdot \gamma \geq H(Y) - \alpha$$

$$T \geq \frac{H(Y) - \alpha - I(Y; \hat{Y}_0)}{p \cdot \gamma}.$$

We argue that the our claim will hold if we can show that $T$ is some finite integer. However, it is easy as the right hand side in the above expression is bounded from above by $\frac{\log K - \alpha}{p \cdot \gamma} \in \mathbb{R}_+$, and hence from the Archimedean property of the real numbers, we can always find a $T \in \mathbb{N}$ for which the property holds, thereby arguing the existence of the iteration step $T$ for which the boosting algorithm achieves the argued $\alpha$-predictive sufficiency. Furthermore, we recall from 6, $I(Y; E \mid \hat{Y}_T) = H(Y \mid \hat{Y}_T) - H(Y \mid \hat{Y}_T, E)$ Without removing the term with $E$, we have $I(Y; E \mid \hat{Y}_T) = H(Y \mid \hat{Y}_T) - H(Y \mid \hat{Y}_T, E) = H(Y) - I(Y; \hat{Y}_T) - H(Y \mid \hat{Y}_T, E)$.

Now to get $I(Y; E \mid \hat{Y}_T) \leq \alpha$, it is sufficient to have $H(Y) - H(Y \mid \hat{Y}_T, E) - \alpha \leq I(Y; \hat{Y}_T)$

$$I(Y; \hat{Y}_0) + Tp\gamma \geq H(Y) - H(Y \mid \hat{Y}_T, E) - \alpha$$

$$T \geq \frac{H(Y) - H(Y \mid \hat{Y}_T, E) - \alpha - I(Y; \hat{Y}_0)}{p \cdot \gamma}$$

To avoid the dependence on $\hat{Y}_T$, we can replace $H(Y \mid \hat{Y}_T, E)$ with its lower bound $H(Y \mid \mathbf{X}, E)$ and still have a valid bound as below.

$$T \geq \frac{H(Y) - H(Y \mid \mathbf{X}, E) - \alpha - I(Y; \hat{Y}_0)}{p \cdot \gamma}$$

$\square$

**Corollary 5.1.** *Under the Assumptions 5.1-5.4, there exists a finite $T < \infty$ such that the predictor $\hat{Y}_t$ learned by Algorithm 1 after $t \geq T$ rounds satisfies $H(U \mid \hat{Y}_t) \leq \beta$ for a small $\beta$. The lower bound on $T$ is given by*

$$T = \frac{H(U) - \beta - c \cdot I(Y; \hat{Y}_0)}{c \cdot p \cdot \gamma}$$

*Proof.* From the proof of the Theorem 5.1, we have the following:

$$I(Y; \hat{Y}_T) \geq I(Y; \hat{Y}_0) + T \cdot p \cdot \gamma \tag{7}$$

From Assumption 5.4, we have the following:

$$I(U; \hat{Y}_T) \geq c \cdot (I(Y; \hat{Y}_0) + T \cdot p \cdot \gamma) \tag{8}$$

$$H(U \mid \hat{Y}_T) = H(U) - I(U; \hat{Y}_T) \leq H(U) - c \cdot (I(Y; \hat{Y}_0) + T \cdot p \cdot \gamma) \tag{9}$$

$$\tag{10}$$

To ensure $H(U \mid \hat{Y}_T) \leq \beta$, it is enough to ensure $H(U) - c \cdot (I(Y; \hat{Y}_0) + T \cdot p \cdot \gamma) \leq \beta$. Solving this for $T$ implies the desired inequality below:

$$T \geq \frac{H(U) - \beta - c \cdot I(Y; \hat{Y}_0)}{c \cdot p \cdot \gamma} \tag{11}$$

$\square$

## A.1 Lower bound on Conditional Informativeness

To prove that $I(Y; \phi(\mathbf{X}) \mid E, \hat{Y})$ is a lower bound on $I(Y; \phi(\mathbf{X}) \mid E)$, we use the key property from the causal graph shown in Figure 1. Since $Y$ and $\hat{Y}$ become independent conditional on $\phi(\mathbf{X})$, we have $I(Y; \hat{Y} \mid \phi(\mathbf{X}), E) = 0$ (conditioning on $E$ does not change the independence property). Now Express the mutual information between $Y$ and the pair $(\phi(\mathbf{X}), \hat{Y})$ given $E$ in two ways:

$$
\begin{aligned}
I(Y; \phi(\mathbf{X}), \hat{Y} \mid E) &= I(Y; \phi(\mathbf{X}) \mid E) + I(Y; \hat{Y} \mid E, \phi(\mathbf{X})), \\
I(Y; \phi(\mathbf{X}), \hat{Y} \mid E) &= I(Y; \hat{Y} \mid E) + I(Y; \phi(\mathbf{X}) \mid E, \hat{Y}).
\end{aligned}
\tag{12}
$$

Equating the right hand side terms of the two equations above and using the fact: $I(Y; \hat{Y} \mid E, \phi(\mathbf{X})) = 0$, we have the following.

$$
\begin{aligned}
I(Y; \phi(\mathbf{X}) \mid E) + 0 &= I(Y; \hat{Y} \mid E) + I(Y; \phi(\mathbf{X}) \mid E, \hat{Y}) \\
I(Y; \phi(\mathbf{X}) \mid E) - I(Y; \phi(\mathbf{X}) \mid E, \hat{Y}) &= I(Y; \hat{Y} \mid E)
\end{aligned}
\tag{13}
$$

Since $I(Y; \hat{Y} \mid E) \geq 0$, we have the desired inequality:

$$
I(Y; \phi(\mathbf{X}) \mid E) \geq I(Y; \phi(\mathbf{X}) \mid E, \hat{Y})
\tag{14}
$$

## B   A Special Case: Gradient Boosting

In this section, we apply our analysis in the main paper to a special type of boosting method called gradient boosting. For ease of understanding, we repeat some of the text from main paper here. Gradient boosting initializes with a base predictor $\hat{Y}_0(\mathbf{X}) = h_0(\mathbf{X})$ (commonly a constant), and at each iteration $t$, it finds a new base learner $h_t(\mathbf{X})$ that best fits the current pseudo-residuals (negative gradients of the loss with respect to the output) and adds $\eta h_t(\mathbf{X})$ to the previous predictor $\hat{Y}_{t-1}(\mathbf{X})$ to get the current predictor $\hat{Y}_t(\mathbf{X}) = \hat{Y}_{t-1} + \eta h_t(\mathbf{X})$, where $\eta$ is the learning rate (see Algorithm 2). We use the notation $R_t((\mathbf{X}), Y)$ to denote the random variable corresponding to the pseudo-residuals at iteration $t$ of the gradient boosting algorithm. To avoid notational clutter, we use $R_t$ and $h_t$ to denote the random variables $R_t(\mathbf{X}, Y)$ and $h_t(\mathbf{X})$.

**Definition B.1** ($\gamma$–approximate weak learner for gradient-boosting). *Let $\hat{Y}_t(\mathbf{X})$ denote the gradient boosting predictor at iteration $t$, and $R_t$ be the random variable corresponding to the pseudo-residual at iteration $t$ with $R_t := -\left.\frac{\partial \ell(Y, \hat{Y}(\mathbf{X}))}{\partial \hat{Y}(\mathbf{X})}\right|_{\hat{Y} = \hat{Y}_{t-1}}$, where $\ell$ is a loss function. Let $\mathcal{H}$ be a class of hypothesis functions $h : \mathcal{X} \to \mathbb{R}$, and let $b(\cdot)$ be a fixed baseline predictor. We say $\mathcal{H}$ satisfies the $\gamma$-MI weak learning condition at iteration $t$ if there exists $h_t \in \mathcal{H}$ such that the following holds for some $\gamma > 0$:*
$$
I(R_t; h_t) \geq I(R_t; b_t(\mathbf{X})) + \gamma
$$
*For a constant baseline $b_t(\cdot)$, $I(R_t; b_t(\mathbf{X})) = 0$ and the condition reduces to $I(R_t; h_t) \geq \gamma$.*

$\gamma$**-approximate weak learners in gradient boosting:** At each iteration $t$, a weak learner is trained to fit the pseudo-residuals $R_t$ using the covariates $\mathbf{X}$ as input. Assumption 5.1 is stated in information-theoretic terms i.e., mutual information between the weak-learner outputs $h_t$ and the residuals $R_t$. It is therefore important to argue that this information-theoretic weak-learning condition holds in gradient boosting. Intuitively, if $R_t$ were independent of $h_t$, then $h_t$ would carry no information about $R_t$ and could not provide a weak-learning edge. In practice, however, the weak learner $h_t$ is obtained by fitting pseudo-residuals and thus typically has nonzero correlation with $R_t$. Nonzero correlation implies statistical dependence and hence strictly positive mutual information $I(R_t; h_t) > 0$. Indeed, because gradient boosting performs functional gradient descent in prediction space at each iteration (Mason et al., 1999), the fitted weak learner $h_t$ is expected to be positively correlated with the pseudo-residuals, i.e., $\text{Cov}(R_t, h_t) > 0$. Nonzero covariance implies statistical dependence, and hence $I(R_t; h_t) > 0$.

Algorithm 1 presents an information-theoretic interpretation of the standard gradient boosting algorithm. For ease of exposition, we omit the usual optimal step-size / line-search computation

---

**Algorithm 2** Standard gradient boosting algorithm. Comments in orange describe the information-theoretic interpretation of the standard gradient boosting algorithm.

---

**Require:** Training set $\{(\mathbf{x}^{(i)}, y^{(i)})\}_{i=1}^n$, loss function $L(y, \hat{y})$, hypothesis class $\mathcal{H}$, learning rate $\eta$, number of boosting iterations $T$.

1: Initialize $\hat{Y}_0(\mathbf{X}) \leftarrow \arg\min_c \sum_{i=1}^n L(y^{(i)}, c)$.
   /* Run for $T$ number of iterations where $T$ can be chosen based on Equation 17 to ensure leaf embeddings are clustered according to hidden confounders. */

2: **for** iteration $t = 1, \ldots, T$ **do**

3:     Compute pseudo-residuals $\{r_t^{(i)}\}_{i=1}^n$ where $r_t^{(i)} = - \left. \frac{\partial L\left(y^{(i)}, \hat{Y}(\mathbf{x}^{(i)})\right)}{\partial \hat{Y}(\mathbf{x}^{(i)})} \right|_{\hat{Y} = \hat{Y}_{t-1}}$.

4:     Find a weak learner $h_t \in \mathcal{H}$ by fitting the weak learner to the pairs $\{(\mathbf{x}^{(i)}, r_t^{(i)})\}_{i=1}^n$ that satisfies $\mathrm{Cov}\left(R_t, h_t\right) > 0$.
   /* $R_t$ is the random variable corresponding to pseudo-residuals. Lemma B.1 shows how a weak learner that fits pseudo-residuals achieves $I(R_t; h_t) \geq \gamma$. */

5:     Update predictor as $\hat{Y}_t(\mathbf{X}) \leftarrow \hat{Y}_{t-1}(\mathbf{X}) + \eta\, h_t(\mathbf{X})$.

6: **end for**

7: return $\hat{Y}_T(\mathbf{X})$

---

performed after Step 4 in Algorithm 2. Under the common assumption that, if the learning rate $\eta$ is sufficiently small, the effect of the omitted step value can be absorbed into $\eta$, so that our theoretical guarantees remain unchanged. We now present lower bounds on $I(R_t; h_t)$ and $I(Y; h_t)$. The lower bound on $I(R_t; h_t)$ provides an explicit $\gamma$ for which Assumption 5.1 holds, while the bound on $I(Y; h_t)$ is needed to prove that gradient boosting returns $\alpha$-predictive sufficient predictors.

**Lemma B.1.** *Let $R_t$ and $h_t$ be the random variables corresponding to residuals and weak learner predictions at iteration $t$ of gradient boosting. Let $g(R_t) : \mathrm{supp}(R_t) \to [0, 1]$ be a measurable function. Assuming $0 < \mathrm{Var}(h_t) \leq V_{\max} < \infty$ and $|\mathrm{Cov}(g(R_t), h_t)| \geq c > 0$, then we have $I(R_t; h_t) \geq \frac{2c^2}{V_{\max}}$.*

*Proof.* Since $g(R_t)$ is a measurable function of $R_t$, we have $I(R_t; h_t) \geq I(g(R_t); h_t)$. Hence, we focus on lower bounding $I(g(R_t); h_t)$ instead. From the properties of mutual information, we have the following:

$$I(g(R_t); h_t) = \mathbb{E}_{h_t}[D_{KL}(\mathbb{P}_{g(R_t)|h_t} \,||\, \mathbb{P}_{g(R_t)})] \tag{15}$$

Where $D_{KL}$ denotes the KL divergence and $\mathbb{P}_{g(R_t)|h_t}$ and $\mathbb{P}_{g(R_t)}$ denote the corresponding probability measures. From Pinsker's inequality, which relates the KL divergence and total variation (TV) distance, we have the following:

$$\begin{aligned} I(g(R_t); h_t) &= \mathbb{E}_{h_t}[D_{KL}(\mathbb{P}_{g(R_t)|h_t} \,||\, \mathbb{P}_{g(R_t)})] \\ &\geq 2\mathbb{E}_{h_t}[TV(\mathbb{P}_{g(R_t)|h_t}, \mathbb{P}_{g(R_t)})^2] \qquad \text{(Applying Pinsker's inequality)} \end{aligned}$$

Since $g(R_t)$ is bounded between $0$ and $1$, from the properties of total variation distance for bounded functions, we can write as follows:

$$I(g(R_t); h_t) \geq 2\mathbb{E}_{h_t}\left[(\mathbb{E}[g(R_t)|h_t] - \mathbb{E}[g(R_t)])^2\right] = 2\,\mathrm{Var}(\mathbb{E}[g(R_t)|h_t]) \tag{16}$$

Now, from Cauchy–Schwarz inequality involving covariance and variances, we have the following:

$$\begin{aligned} \mathrm{Cov}(\mathbb{E}[g(R_t) \mid h_t], h_t)^2 &\leq \mathrm{Var}(\mathbb{E}[g(R_t) \mid h_t])\, \mathrm{Var}(h_t) \\ \mathrm{Var}(\mathbb{E}[g(R_t) \mid h_t]) &\geq \frac{\mathrm{Cov}(\mathbb{E}[g(R_t) \mid h_t], h_t)^2}{V_{\max}} \\ &= \frac{\mathrm{Cov}(g(R_t), h_t)^2}{V_{\max}} \\ &= \frac{c^2}{V_{\max}} \end{aligned}$$

Finally, from Equation equation 16 and the first line of the proof we have $I(R_t; h_t) \geq I(g(R_t); h_t) \geq \frac{2c^2}{V_{\max}}$. $\qquad\square$

Lemma B.1 shows that the weak learner $h_t$ provides information about the pseudo-residual $R_t$. We next show that $h_t$ also carries nontrivial information about the label $Y$.

**Theorem B.2.** *Let $R_t, h_t$ denote the random variables corresponding to residuals and weak learner predictions at iteration $t$ of gradient boosting. Let $g(Y) : \mathcal{Y} \to [0, 1]^m$ be a measurable function. Let $s(\mathbf{X}) = \mathbb{E}[R_t \mid \mathbf{X}]$ and $\eta(\mathbf{X}) = \mathbb{E}[g(Y) \mid \mathbf{X}]$. Assume $h_t$ satisfies the standard weak learning condition $\mathbb{E}[R_t h_t] \geq \rho > 0$, and assume local linearity of the expected gradient, i.e., $s(\mathbf{X}) = a^T(\eta(\mathbf{X}) - \bar{\eta}) + k(\mathbf{X})$ for some $a \in \mathbb{R}^m; ||a|| = \lambda > 0$ and for some $k$. Let $\bar{\eta} = \mathbb{E}[\mathbb{E}[g(Y) \mid \mathbf{X}]]$ and either $k(\mathbf{X}) = 0$ or $\mathbb{E}[k(\mathbf{X})h_t] = 0$. Additionally, assume $\mathrm{Var}(h_t) \leq V_{max} < \infty$. Then, we have : $I(Y; h_t) \geq \frac{2\rho^2}{m\lambda^2 V_{max}}$*

*Proof.* From the weak learning assumption about $h_t$ i.e., $\mathbb{E}[R_t h_t] \geq \rho > 0$, we have the following

$$\mathbb{E}[R_t h_t] = \mathbb{E}[\mathbb{E}[R_t \mid \mathbf{X}]h_t] = \mathbb{E}[s(\mathbf{X})h_t] \geq \rho$$

From the assumption of the linearity of the expected gradient i.e., $s(\mathbf{X}) = a^T(\eta(\mathbf{X}) - \bar{\eta}) + k(\mathbf{X})$, we have the following:

$$\mathbb{E}[s(\mathbf{X})h_t] = a^T\mathbb{E}[(\eta(\mathbf{X}) - \bar{\eta})h_t] + \mathbb{E}[k(\mathbf{X})h_t] = a^T\mathbb{E}[(\eta(\mathbf{X}) - \bar{\eta})h_t] \geq \rho$$

If we set $v = \mathbb{E}[(\eta(\mathbf{X}) - \bar{\eta})h_t]$, from Cauchy-Schwarz inequality, we have the following:

$$|a^T v| \leq ||a||_2 ||v||_2 \implies ||v||_2 \geq \frac{\rho}{\lambda}$$

Since $||v||_2 \geq \frac{\rho}{\lambda}$, then there is atleast one coordinate $v_q$ of $v \in \mathbb{R}^m$ that satisfies the following:

$$|v_q| \geq \frac{||v||_2}{\sqrt{m}} \geq \frac{\rho}{\lambda\sqrt{m}}$$

Since $v_q = \mathbb{E}[(\eta_q(\mathbf{X}) - \bar{\eta}_q)h_t]$, using $\eta_q = \mathbb{E}[g_q(Y) \mid \mathbf{X}]$ and $\bar{\eta}_q = \mathbb{E}[g_q(Y)]$, we have the following:

$$v_q = \mathbb{E}[\mathbb{E}[g_q(Y) \mid \mathbf{X}]h_t] - \mathbb{E}[g_q(Y)]\mathbb{E}[h_t] = \mathbb{E}[g_q(Y)h_t] - \mathbb{E}[g_q(Y)]\mathbb{E}[h_t] = \mathrm{Cov}(g_q(Y), h_t)$$

From the Lemma B.1, the term $c$. Following the similar arguments from the previous proof with $c \geq \frac{\rho}{\lambda m}$, we have the desired inequality: $I(Y; h_t) \geq I(g_q(Y); h_t) \geq \frac{2(\rho/\lambda\sqrt{m})^2}{V_{\max}} = \frac{2\rho^2}{m\lambda^2 V_{max}}$

$\square$

While the exact bound in Theorem B.2 is not crucial for our analysis, the theorem establishes that each weak learner is individually predictive of the label $Y$, i.e., $I(Y; h_t) > 0$. In the following, we make a conditional weak-learning assumption (Assumption 5.2) ensuring that each weak learner $h_t$ contributes a nontrivial amount of *new predictive information* beyond what is already captured by the previous weak learners $h_0, \ldots, h_{t-1}$.

**Assumption B.1** (Conditional $\gamma$-approximate weak learning). *We assume that there exists a constant $\gamma > 0$ such that for all iterations $t$, $I(Y; h_t \mid h_0, \ldots, h_{t-1}) \geq \gamma$.*

**Assumption B.2** (Strong learner is a deterministic function of weak learners). *At any iteration $t$, we assume that $\hat{Y}_t$ is a deterministic function of $(h_0, \ldots, h_t)$.*

Assumption B.2 is required to decompose the predictive information $I(Y; \hat{Y})$ into contributions from each weak individual learner. In practice, this assumption holds as the models are usually deterministic once they are trained. To ensure that the information present in each new weak learner is retained by the additive aggregation (i.e., the final prediction formed as a weighted sum of weak learners), we make the following assumption.

**Assumption B.3** (Monotone information gain). *At any iteration $t$ of the gradient boosting algorithm, if $h_t$ satisfies the conditional $\gamma$-approximate weak learning condition, we assume that there exists a $\delta_t \in [0, \gamma)$, such that $I(Y; \hat{Y}_t) \geq I(Y; \hat{Y}_{t-1}) + I(Y; h_t \mid h_0, \ldots, h_{t-1}) - \delta_t$. Additionally, we assume that for all $T$, $\sum_{t=0}^{T} \delta_t \leq \Delta$*

We now show that Algorithm 1 returns an $\alpha$-predictive sufficient predictor after a finite number of boosting iterations.

**Theorem B.3.** *Under Assumptions B.1, 5.3, and B.3, there exists a finite $T < \infty$ such that the predictor $\hat{Y}_t$ learned by Algorithm 2 after $t \geq T$ iterations is $\alpha$-predictive sufficient. The lower bound $T$ is given by*

$$T = \frac{H(Y) - H(Y \mid \mathbf{X}, E) - \alpha - I(Y; \hat{Y}_0) + \Delta}{\gamma} \tag{17}$$

*When $E$ is unknown, the same result holds with a conservative upper bound obtained by setting $H(Y \mid \mathbf{X}, E) = 0$.*

*Proof.* From Assumption 5.3, $\hat{Y}_T$ is a deterministic function of $(h_0, h_1, h_2, \ldots, h_T)$, and $I(Y; \hat{Y}_0) = I(Y; h_0)$ because $\hat{Y}_0$ is initialized to $h_0$ before boosting training loop. From Assumption B.3, we have the following:

$$I(Y; \hat{Y}_T) \geq I(Y; \hat{Y}_{T-1}) + I(Y; h_T \mid h_0, \ldots, h_{T-1}) - \delta_T \tag{18}$$

Consider any iteration $t \geq 1$, Then, there exists a gamma $\gamma$ such that the following holds.

$$I(Y; h_T \mid h_0, \ldots, h_{T-1}) \geq \gamma \tag{19}$$

From Equations (18) and (19) we have the following.

$$I(Y; \hat{Y}_T) \geq I(Y; \hat{Y}_{T-1}) + \gamma - \delta_T \tag{20}$$

Applying the Assumption B.3 repeatedly for all $t$, we obtain the following.

$$\begin{aligned} I(Y; \hat{Y}_T) &\geq I(Y; \hat{Y}_0) + T\gamma - \sum_{t=1}^{T} \delta_t \\ &\geq I(Y; \hat{Y}_0) + T\gamma - \Delta \end{aligned} \tag{21}$$

Given $\hat{Y}$, $Y - \hat{Y}$ and $Y$ have one-to-one correspondence. Hence, it follows that: $I(Y - \hat{Y}; E \mid \hat{Y}) = I(Y; E \mid \hat{Y})$. We next use the following simple upper bound on $I(Y; E \mid \hat{Y}_T)$:

$$I(Y; E \mid \hat{Y}_T) = H(Y \mid \hat{Y}_T) - H(Y \mid \hat{Y}_T, E) \leq H(Y \mid \hat{Y}_T), \tag{22}$$

and hence $I(Y; E \mid \hat{Y}_T) \leq H(Y \mid \hat{Y}_T)$. Furthermore, since $H(Y \mid \hat{Y}_T) = H(Y) - I(Y; \hat{Y}_T)$, we have: $I(Y; E \mid \hat{Y}_T) \leq H(Y) - I(Y; \hat{Y}_T)$. Now to get $I(Y; E \mid \hat{Y}_T) \leq \alpha$, from Equation equation 21, it is sufficient to enforce $H(Y) - \alpha \leq I(Y; \hat{Y}_0) + T\gamma - \Delta$ as follows.

$$\begin{aligned} I(Y; \hat{Y}_0) + T\gamma - \Delta &\geq H(Y) - \alpha \\ T &\geq \frac{H(Y) - \alpha - I(Y; \hat{Y}_0) + \Delta}{\gamma} \end{aligned} \tag{23}$$

Now, from 22, $I(Y; E \mid \hat{Y}_T) = H(Y \mid \hat{Y}_T) - H(Y \mid \hat{Y}_T, E)$ Without removing the term with $E$, we have $I(Y; E \mid \hat{Y}_T) = H(Y \mid \hat{Y}_T) - H(Y \mid \hat{Y}_T, E) = H(Y) - I(Y; \hat{Y}_T) - H(Y \mid \hat{Y}_T, E)$.

Now to get $I(Y; E \mid \hat{Y}_T) \leq \alpha$, it is sufficient to have $H(Y) - H(Y \mid \hat{Y}_T, E) - \alpha \leq I(Y; \hat{Y}_T)$

$$\begin{aligned} I(Y; \hat{Y}_0) + T\gamma - \Delta &\geq H(Y) - H(Y \mid \hat{Y}_T, E) - \alpha \\ T &\geq \frac{H(Y) - H(Y \mid \hat{Y}_T, E) - \alpha - I(Y; \hat{Y}_0) + \Delta}{\gamma} \end{aligned}$$

To avoid the dependence on $\hat{Y}_T$, we can replace $H(Y \mid \hat{Y}_T, E)$ with its lower bound $H(Y \mid \mathbf{X}, E)$ and still have a valid bound as below.

$$T \geq \frac{H(Y) - H(Y \mid \mathbf{X}, E) - \alpha - I(Y; \hat{Y}_0) + \Delta}{\gamma} \tag{24}$$

$\square$

Theorem B.3 guarantees that the gradient boosting algorithm converges in finite number of iterations, and lead to the $\alpha$-sufficient predictor. We now prove the correspondence between a gradient boosting model's leaf embeddings and the hidden confounder variable $U$. Since leaf embeddings in gradient boosting correspond to both outcome and input representations, we show how boosting achieves $H(U \mid \hat{Y}_t) \leq \beta$ for some $\beta \geq 0$ such that after the time step $t$, the uncertainty in $U$ given the predictions $\hat{Y}_t$ is less than or equal to $\beta$. We start with the assumption below.

Motivated by the causal graph in Figure 1, Assumption 5.4 requires that $\phi(\mathbf{X})$ and the learning procedure preserve the components of $\mathbf{X}$ that carry $U$-signal, and the predictor actually uses those components rather than discarding them.

**Corollary B.1.** *Under Assumptions 5.2, 5.3, B.3, and 5.4, there exists a finite $T < \infty$ such that the predictor $\hat{Y}_t$ learned by Algorithm 2 after $t \geq T$ iterations satisfies $H(U \mid \hat{Y}_t) \leq \beta$. The lower bound $T$ is given by*

$$T = \frac{H(U) - \beta - cI(Y; \hat{Y}_0) + c\Delta}{c\gamma} \tag{25}$$

*Proof.* From the proof of the Theorem 5.1, we have the following:

$$I(Y; \hat{Y}_T) \geq I(Y; \hat{Y}_0) + T\gamma - \Delta \tag{26}$$

From Assumption 5.4, we have the following:

$$I(U; \hat{Y}_T) \geq c \cdot (I(Y; \hat{Y}_0) + T\gamma - \Delta) \tag{27}$$

$$H(U \mid \hat{Y}_T) = H(U) - I(U; \hat{Y}_T) \leq H(U) - c \cdot (I(Y; \hat{Y}_0) + T\gamma - \Delta) \tag{28}$$

$$\tag{29}$$

To ensure $H(U \mid \hat{Y}_T) \leq \beta$, it is enough to ensure $H(U) - c \cdot (I(Y; \hat{Y}_0) + T\gamma - \Delta) \leq \beta$. Solving this for $T$ implies the desired inequality below:

$$T \geq \frac{H(U) - \beta - cI(Y; \hat{Y}_0) + c\Delta}{c\gamma} \tag{30}$$

$\square$

We now discuss how to choose a finite $T$ based on the theorem above. Assuming $\gamma > 0$, the right-hand sides of Equation equation 23 and Equation equation 24 are bounded above by $\frac{\log K - \alpha + \Delta}{\gamma}$ (for discrete $Y$ with $K$ possible values), and by $\frac{\frac{1}{2}\log(2\pi e\sigma^2) - \alpha + \Delta}{\gamma}$ (for continuous $Y$ with variance bounded by $\sigma^2$), where the continuous bound follows from the fact that a Gaussian maximizes differential entropy at fixed variance. Let $R$ denote the appropriate upper bound (one of the two expressions above). If $R \leq 0$ then any $T \in \mathbb{N}$ suffices; otherwise choose $T = \lceil R \rceil \in \mathbb{N}$ By construction $T$ is finite and satisfies the inequality in equation 17, which establishes the existence of an iteration $T$ after which the boosting algorithm achieves the claimed $\alpha$-predictive sufficiency.

## C  ADDITIONAL EXPERIMENTAL RESULTS

Figures C1- C4 show the results with respect to various choices of hyperparameters, dimensionality reduction methods, and metrics. A key takeaway from these results is that better clustering with respect to hidden confounders is consistently associated with better performance. This explains the reason behind the success of boosting methods. Figure C5 shows ID and OOD MSE for different values of distribution shifts. Since high distribution shifts cannot satisfy the common confounder support assumption, the generalization performance drops significantly due to large shifts in data distribution. Figure C5 shows performance comparison of XGBoost and CatBoost with invariant risk minimization (IRM) (Arjovsky et al., 2019) and group DRO (Sagawa et al., 2019). We consider the same setup as the synthetic experiment 2 presented in the main paper, where the causal graph is: $U_1 \rightarrow X, U_1 \rightarrow Y, X \rightarrow Y, U_2 \rightarrow S, U_2 \rightarrow Y, U_1 \sim \mathcal{N}(\mu_e, \sigma_e), U_2 \sim \mathcal{N}(\mu_f, \sigma_f)$. When only $X$ is used as input, we observe that XGBoost and CatBoost perform better than IRM and GroupDRO,

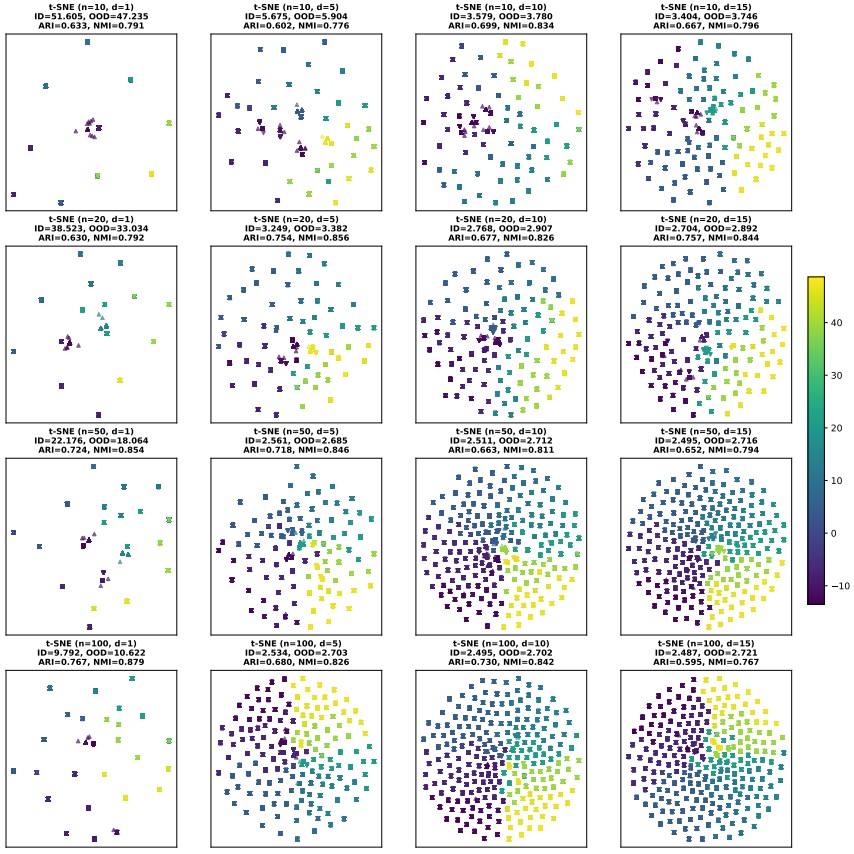

Figure C1: Results on CatBoost method. Dimensionality reduction is done using t-SNE. Results show the goodness of clustering with respect to the hidden confounder value. Above each subplot, clustering metrics: ARI, NMI are reported along with hyperparameter choices, ID, and OOD MSE values.

indicating that boosting methods are effective under a hidden confounding shift. When we use both $X, S$ as inputs, only GroupDRO performs on par with boosting, while IRM still performs worse. This also shows that invariance learning is insufficient for generalization under a hidden confounding shift.

**Real-world Experiments:** We conduct experiments on five large-scale tabular OOD benchmark datasets for binary classification: *Readmission, Foodstamps, Diabetes, Income*, and *Unemployment*. The datasets and their domain splits are taken from the TableShift benchmark (Gardner et al., 2023). Prior work shows these datasets exhibit hidden confounding when evaluated via observed distribution shifts involving $\mathbf{X}, Y$ and $\phi(\mathbf{X})$ (Gardner et al., 2023; Reddy et al., 2026); therefore, the environment variables $E$ provided with these datasets are appropriate for assessing our proposed $\alpha$-predictive sufficiency. Note that we do not use $E$ for model training. We show average results computed over 5 random seeds.

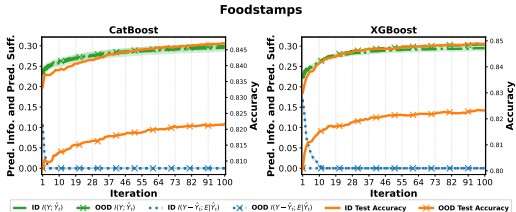

Figure C6: Comparison of accuracy, predictive information, and predictive sufficiency on the Foodstamps dataset.

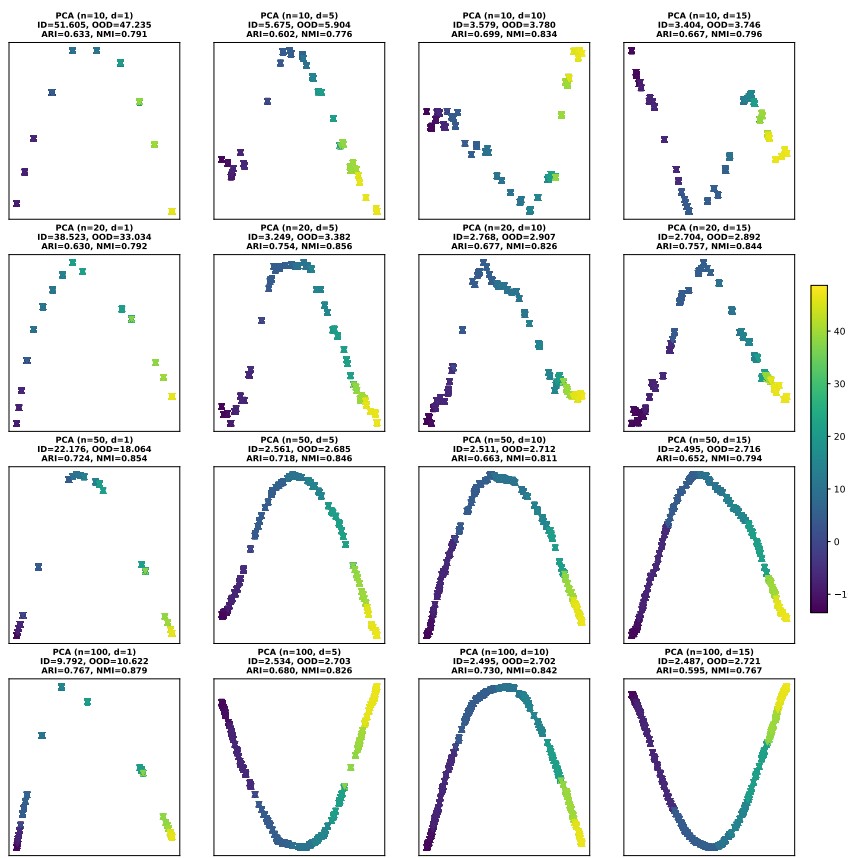

Figure C2: Results on CatBoost method. Dimensionality reduction is done using PCA. Results show the goodness of clustering with respect to the hidden confounder value. Above each subplot, clustering metrics: ARI, NMI are reported along with hyperparameter choices, ID, and OOD MSE values.

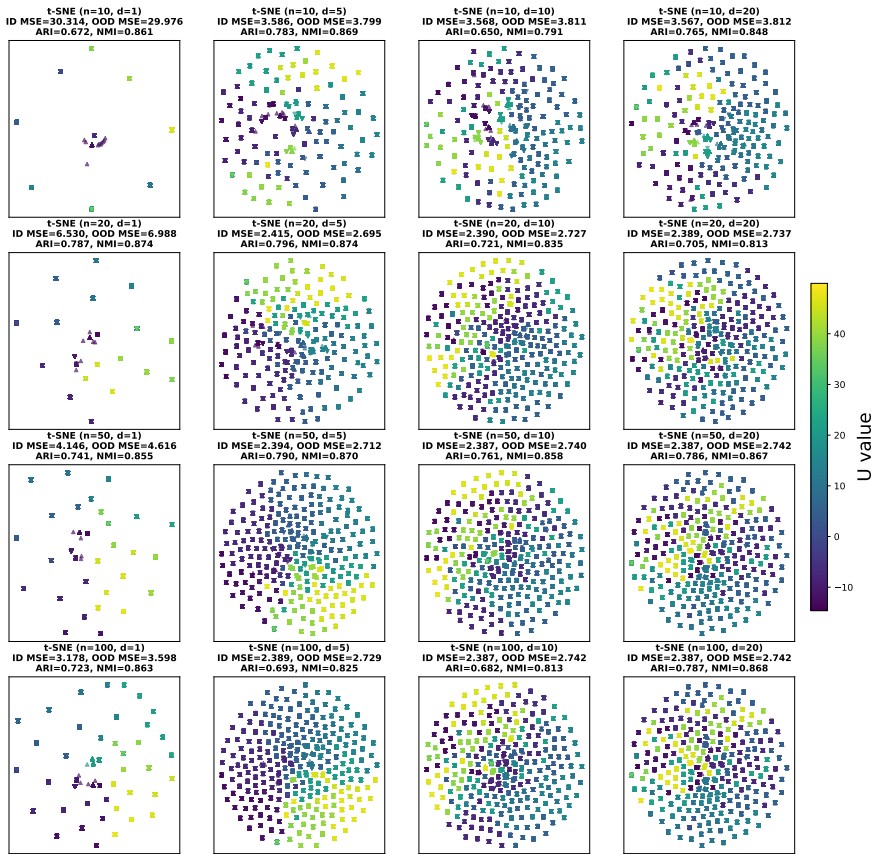

Figure C3: Results on XGBoost method. Dimensionality reduction is done using t-SNE. Results show the goodness of clustering with respect to the hidden confounder value. Above each subplot, clustering metrics: ARI, NMI are reported along with hyperparameter choices, ID, and OOD MSE values.

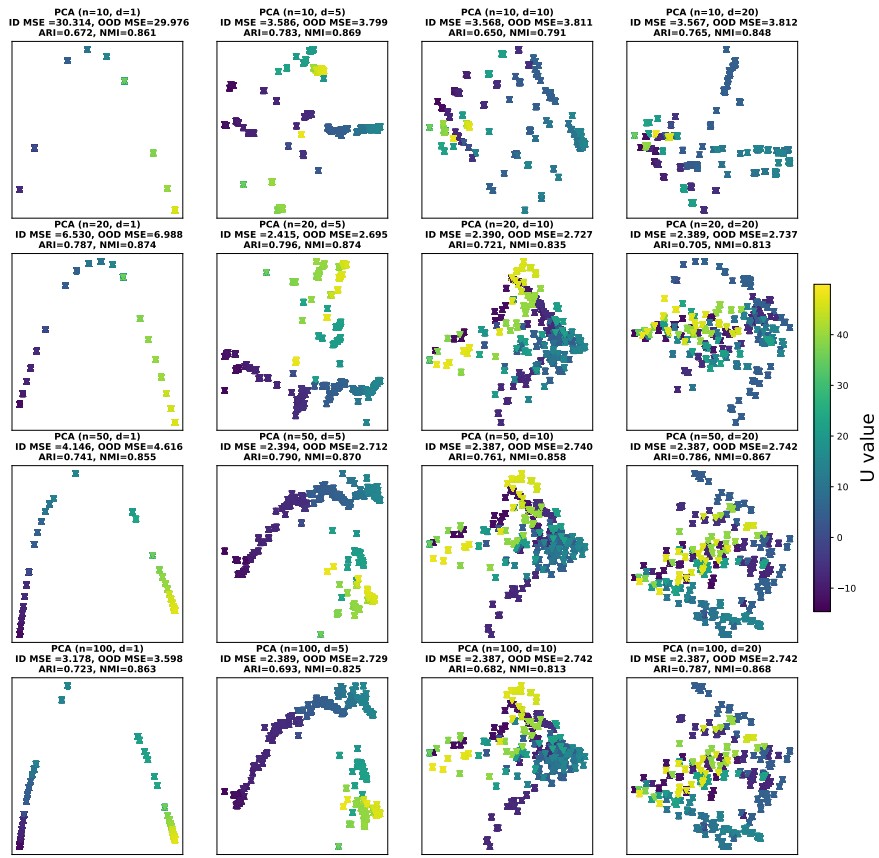

Figure C4: Results on XGBoost method. Dimensionality reduction is done using PCA. Results show the goodness of clustering with respect to the hidden confounder value. Above each subplot, clustering metrics: ARI, NMI are reported along with hyperparameter choices, ID, and OOD MSE values.

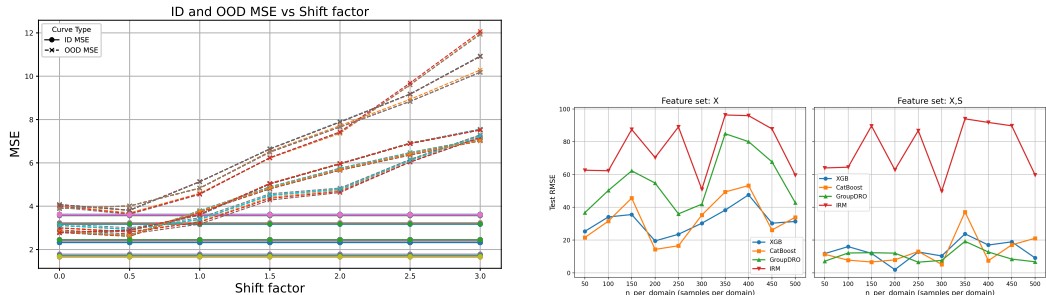

Figure C5: left: Performance of XGBoost for different shift factor values. Colors indicate different combinations of *number of trees, number of samples in each domain, and depths of trees.* right: Comparison with OOD generalization methods.

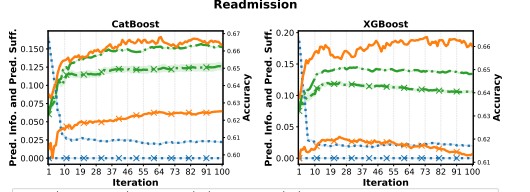
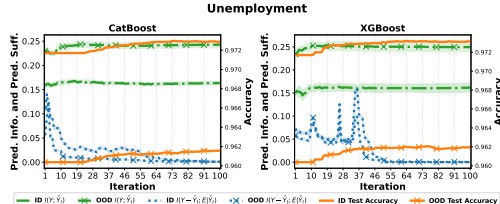

Figure C7: Comparison of accuracy, predictive information, and predictive sufficiency on Diabetes and Income datasets.

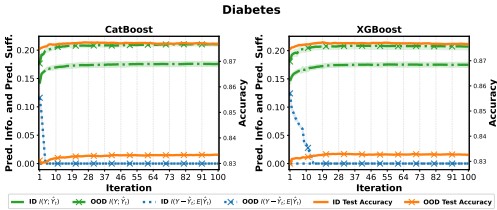
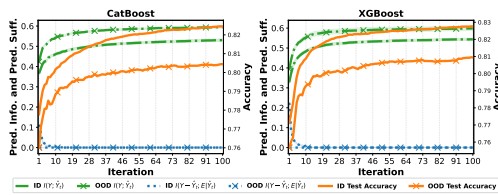

Figure C8: Comparison of accuracy, predictive information, and predictive sufficiency on Diabetes and Income datasets.

From Figure C7, we observe that the predictive information $I(Y; \hat{Y}_t)$ for both ID and OOD test data increases as the number of trees/iterations $t$ increases. Crucially, $I(Y - \hat{Y}_t; E \mid \hat{Y}_t)$ decreases toward zero as $t$ increases. Thus, even though $E$ is not used explicitly during training, gradient-boosting methods empirically achieve predictive sufficiency: the prediction errors $Y - \hat{Y}_t$ become independent of the environment variables induced by hidden confounders as the model performance converges. From the results, we observe that $I(Y - \hat{Y}_t; E \mid \hat{Y}_t)$ remains zero for some datasets because the available data contain only a single environment. Since $E$ is constant in this setting, the conditional mutual information is trivially zero.

Figures C6 and C8 extend the real-world experiments from the main paper. Here we show results on Foodstamps, Diabetes, and Income datasets. Similar to the main paper results, we observe that the predictive information $I(Y; \hat{Y}_t)$ for both ID and OOD test data increases as the ensemble grows (i.e., as the number of trees / iterations $t$ increases). Crucially, $I(Y - \hat{Y}_t; E \mid \hat{Y}_t)$ decreases toward zero as $t$ increases. Thus, even though $E$ is not used explicitly during training, gradient-boosting methods empirically achieve predictive sufficiency: the prediction errors $Y - \hat{Y}_t$ become (approximately) independent of the environment variables induced by hidden confounders as the model converges. From the results, we observe that $I(Y - \hat{Y}_t; E \mid \hat{Y}_t)$ remains zero for some datasets because the available data contain only a single environment. Since $E$ is constant in this setting, the conditional mutual information is trivially zero.

