# OpenReview forum: "Boosting for Predictive Sufficiency"
_ICLR.cc/2026/Conference — ICLR 2026 Poster_

### Official Review · Reviewer_75bL · 2025-10-21

**Soundness:** 2
**Presentation:** 2
**Contribution:** 2
**Rating:** 2
**Confidence:** 4

**Summary:**

The paper introduces an information theoretic notion called $\alpha$-predictive sufficiency to measure the generalizatiability of predictive models under hiding confounding shifts. It shows that boosting algorithms gives a predictor that is $\alpha$-predictive sufficient and they implicitly identifies environments corresponding to hidden confounding shifts.

**Strengths:**

- The paper is mostly well-written. The introduction gives a very clear picture what problems it is trying to solve and sufficient background information, including the relevant work.

- The observation on achieving $\alpha$-predictive sufficiency by maximizing the predictive information is insightful. Using $\alpha$-predictive sufficiency to evaluate the invariance of the predictors is logically sound given that equation 1 holds.

- Proposition 4.1 seems correct.

- The paper provides codes to reproduce the experimental results.

**Weaknesses:**

- The paper acknowledges that the idea of $\alpha$-predictive sufficiency is inspired by by $\alpha$-approximate multi-calibration. It would be better to include some high-level descriptions about $\alpha$-approximate multi-calibration in the background. Otherwise, the flow of the paper in lines 282-285 becomes a bit disconnected for someone who does not know anything about $\alpha$-approximate multi-calibration.

- Definition 5.1 is not quite clear.  For example, it is not clear why definition 5.1 says ‘then, there exists…’ and it also seems to use $c_{E}$ and $c_{G}$ interchangeably. The subscript $G$ does not seem to carry any meaning.

-  The main claim in Theorem 5.1 is built on the assumptions 5.1-5.3, but the description of the assumptions 5.1 is not clear enough for readers to assess the significance of Theorem 5.1.

- The derivation of the predictive information (Eq. 1) based on a non-peer-reviewed paper seems very questionable. It is built on assumptions that are not described in the submitted manuscript about the information flow between variables.

- The main theorem 5.1 is heavily built on equation 1 in the paper, but equation 1 is derived by Gowtham Reddy et al. 2025. The contribution of the paper seems limited by simply adding assumptions 5.1-5.3 for showing boosting algorithms to be $\alpha$-predictive sufficient.

- Some steps in the proof of Theorem 5.1 are not clear.

- The instructions in the code says to reproduce Figure 2 by running ‘CatBoost_Cluster_wrt_U.py’, but it doesn’t use 100 estimators as shown in the paper.  It only shows up to 40 estimators with max_depth =5.

- In the synthetic experiment 2, the paper does not explain why the boosting algorithm will need $S$ to improve its performance.

- The real-world data seems synthetic by simulating an artificial hidden confounding shift.

**Questions:**

- Why does the mutual information of the residual and the environment variable given the prediction need to be upper bounded by 1 for $\alpha$-predictive sufficiency in Definition 4.2?
- What is $c_{E}$ in definition 5.1? Is $c_{G}$ supposed to be written as $c_{E}$?
- What the set of covariates $\mathbf{X}$ is defined as the elements of environment $E$ in definition 5.1? How should one interpret it as they are separate nodes in the causal graph in Figure 1?
- Is $p$ positive in Assumption 5.1?
- Can the authors explain why assumption 5.1 is realistic?
- For the derivation of equation 1, it relies on the two equations (eq 2 and eq 3) in Gowtham Reddy et al. 2025. The derivation of those inequalities claims that by conditioning on Y, the mutual information between $\phi(X)$ and $E$ will increase, and it will be greater than or equal to $I(\phi(X); E)$, but there is no theoretical support for this claim. A similar issue occurs for other inequalities in eq 2 and eq 3. Can the authors justify these claims to demonstrate the correctness of eq 1 in the submitted manuscript?
- How is line 747 implied by Equation 4 in the proof in Theorem 5.1?
- Why did the experiment choose to shift the mean by 0.2 only in the synthetic experiment 1 as indicated by the code? Can the experiment show the effect of the $\mu_{e}$ for a wider range of values to see how robust the boosting algorithms are?
- How is the shift implemented in the synthetic experiment 2? What is the role of $S$ in this experiment? Why would XGboost fail even when it is only trained by $X$? Isn’t that the opposite of what the paper claims when the hidden confounder shifts?

---

> ### Author Response · Authors · 2025-11-21
> **Response to Reviewer 75bL**
>
> Thank you for your insightful review. Below we address each of your comments.
>
> > What is $c_E$ in definition 5.1? Is $c_G$ supposed to be written as $c_E$? Definition 5.1 is not quite clear. For example, it is not clear why definition 5.1 says ‘then, there exists…’ and it also seems to use $c_E$ and $c_G$ interchangeably. The subscript  does not seem to carry any meaning.
>
> $c_E$ refers to the baseline predictor in an environment $E$. We thank you for pointing out the typo. In the revised version, we have corrected the typo by replacing $c_G$ with $c_e$ (after introducing a new notation $e$ to avoid overloading the notation $E$). We have also changed Definition 5.1 to an assumption and updated the narrative accordingly.
>
> > The derivation of the predictive information (Eq. 1) based on a non-peer-reviewed paper seems very questionable. It is built on assumptions that are not described in the submitted manuscript about the information flow between variables.
>
> Equation (1) follows directly from the causal graph in Figure 1. Although the expression is borrowed from a preprint, its derivation is correct and reliable, as it directly follows from the rules of mutual information and causal information flow. We use Equation (1) solely to motivate the connection to predictive sufficiency. Our theory is otherwise self-contained.
>
> > The main theorem 5.1 is heavily built on equation 1 in the paper, but equation 1 is derived by Gowtham Reddy et al. 2025. The contribution of the paper seems limited by simply adding assumptions 5.1-5.3 for showing boosting algorithms to be $\alpha$-predictive sufficient.
>
> Our main contributions include the novel interpretation of boosting from an information theoretic perspective and connecting boosting to $\alpha$-predictive sufficiency, a novel concept that we have introduced (see Proposition 4.1 and Theorem 5.1). Equation (1) is only the starting point of our analysis. During the rebuttal, we have also added a new corollary (Corollary 5.1) based on a realistic additional assumption (Assumption 5.5, that says $I(U;\hat{Y}) \geq c \cdot I(Y;\hat{Y})$ for some $c>0$.) that shows how the entropy of a hidden confounder reduces after observing outcomes. This strengthens the earlier theorem that boosting returns $\alpha$-sufficient predictors.
>
> We are the first to theoretically explain how boosting can return $\alpha$-sufficient predictors that are better than multicalibrated predictors.
> An important consequence of this are novel insights into boosting and why it tends to outperform other methods on tabular datasets. Prior work explains boosting’s strengths with variance reduction, handling missing variables, feature selection, and yielding multi-calibrated predictors. We show that a core strength of boosting is $\alpha$-sufficiency and thus high robustness to confounding shifts.
>
>
> > The instructions in the code says to reproduce Figure 2 by running ‘CatBoost_Cluster_wrt_U.py’, but it doesn’t use 100 estimators as shown in the paper. It only shows up to 40 estimators with max_depth =5.
>
> We confirm that we used 100 estimators for the results shown in the paper. We believe the setting of 40 estimators with max_depth=5 must have been used for development purposes.
>
> > The real-world data seems synthetic by simulating an artificial hidden confounding shift.
>
> We only change the way train and test datasets are constructed from the pool of data points. Many real-world datasets come with pre-defined train and test splits. In this work, to ensure common confounding support (Assumption 3.2), we only change the train and test split.
>
> > Why does the mutual information of the residual and the environment variable given the prediction need to be upper bounded by 1 for $\alpha$-predictive sufficiency in Definition 4.2?
>
> In the revised version, we have modified the definition to say that $\alpha\geq 0$ instead of $\alpha\in [0,1]$.
>
> > What the set of covariates $\mathbf{X}$ is defined as the elements of environment $E$ in definition 5.1? How should one interpret it as they are separate nodes in the causal graph in Figure 1?
>
> We have changed the notation to avoid the overloading of the notation $E$. In the revised version, we have two separate notations for environment variable and the subset of covariates that form a specific environment. With this change, this confusion gets resolved.
>
> > Is $p$ positive in Assumption 5.1?
>
> Yes, $p\in(0,1)$. We have clarified this point in Assumption 5.1 (Assumption 5.2 in the revised version).

---

> ### Author Response · Authors · 2025-11-21
> **Response to Reviewer 75bL Continued**
>
> > The main claim in Theorem 5.1 is built on the assumptions 5.1-5.3, but the description of the assumptions 5.1 is not clear ...
>
> Asumption 5.1 simply says that at each training iteration of a boosting algorithm, it should be possible to identify a reweighting of training data for which a weak learner can be obtained. For instance, in level-set boosting [1], each training iteration identifies the subset of data for which $\hat{Y}_t = v$ for some $v$ and obtains a weak learner that performs optimally on that subset. Assumption 5.1 is even more general than the level-set boosting argument made above. That is, it is possible to consider the entire dataset with more weight given to the data points with $\hat{Y}_t = v$ instead of selecting a strict subset of the data.
>
>
> [1] Globus-Harris, Ira, et al. "Multicalibration as Boosting for Regression." International Conference on Machine Learning. PMLR, 2023.
>
> > For the derivation of equation 1, it relies on the two equations (eq 2 and eq 3) in Gowtham Reddy et al. 2025. The derivation of those inequalities claims that by conditioning on Y, the mutual information between $\phi(X)$ and ...?
>
> In our paper, we consider the case where $\mathbf{X}\to Y$. So we focus on the inequalities in (2) from [1], because these inequalities are enough to obtain Equation (1) in our paper. We will now explain (i) how the mutual information between $\phi(\mathbf{X}), E$ is affected with and without conditioning on $Y$ and (ii) how the mutual information between $Y, E$ is affected with and without conditioning on $\phi(\mathbf{X})$. We follow the causal graph from Figure (1).
>
> Consider the mutual information between $\phi(X)$ and $E$. There are two paths between $\phi(\mathbf{X})$ and $E$ : (a) $\phi(\mathbf{X})\leftarrow \mathbf{X} \leftarrow U \leftrightarrow E$ and (b) $\phi(\mathbf{X})\leftarrow \mathbf{X} \rightarrow  Y \leftarrow U \leftrightarrow E$. Without conditioning on $Y$, information between $\phi(\mathbf{X})$ and $E$ flows only via the path (a) because $Y$ in path (b) is a collider. However, conditioning on $Y$ opens the path $\phi(\mathbf{X})\leftarrow \mathbf{X}\rightarrow Y \leftarrow U\leftarrow E$ from $\phi(\mathbf{X})$ to $E$ at the collider node $Y$, which results in additional information flow between $\phi(\mathbf{X})$ and $E$ via $Y$. This leads to higher mutual information between $\phi(X)$ and $E$, when conditioned on $Y$.
>
> Now consider the mutual information between $Y$ and $E$. There are two paths between $Y$ and $E$: (a) $Y \leftarrow U \leftrightarrow E$ and (b) $Y  \leftarrow \mathbf{X} \leftarrow U \leftrightarrow E$. Without conditioning on $\phi(\mathbf{X})$, information between $Y$ and $E$ flows through both the paths (a) and (b). However, conditioning on $\phi(\mathbf{X})$ partially blocks the information flow from $Y$ to $E$ at the node $\mathbf{X}$ assuming $\phi(\mathbf{X})$ encodes some information about $\mathbf{X}$. This leads to lower mutual information between $Y$ and $E$ when conditioned on $\phi(\mathbf{X})$.
>
> References:
>
> [1] Abbavaram et al. When shift happens-confounding is to blame. arXiv preprint arXiv:2505.21422, 2025.
>
> > Some steps in the proof of Theorem 5.1 are not clear. How is line 747 implied by Equation 4 in the proof in Theorem 5.1?
>
> Apologies for the confusion. The earlier proof omitted the term $I(Y;\hat{Y}_0)$ due to a typo in which $I(Y;h_1,\dots,h_t)$ was written in place of $I(Y;h_0,\dots,h_t)$. We have corrected this error in the revised manuscript and clarified the proof steps. With these changes, we believe the proof is clearer and easier to follow.
>
>
> > Why did the experiment choose to shift the mean by 0.2 only in the synthetic experiment ..?
>
> It is crucial to satisfy the common confounding support assumption (Assumption 3.2) for reliable generalization. When the hidden confounder distribution shifts substantially between the training and test environments, this assumption is violated, leading to degraded generalization performance. To further illustrate this, we conducted additional experiments with varying shift factor values. As the shift factor increases, the hidden confounder distribution diverges more sharply, and we observe a corresponding drop in OOD performance. These new experimental results are included in Figure B5 of the appendix of the revised paper.

---

> ### Author Response · Authors · 2025-11-21
> **Response to Reviewer 75bL Continued**
>
> > How is the shift implemented in the synthetic experiment 2? What is the role of $S$ in this experiment? ...
>
>
> The shift between training and test environments is implemented by shifting the values of the hidden confounder values. In the code, this is controlled by the *shift_factor* variable. In synthetic experiment 2, the outcome $Y$ is caused by two confounders $U_1$ and $U_2$, where $X$ is caused by $U_1$ and $S$ is caused by $U_2$. When only $X$ is observed, the model cannot recover clustering with respect to both $U_1$ and $U_2$. Only after observing $S$ along with $X$ is the model able to cluster its representations with respect to both $U_1$ and $U_2$. This does not contradict the claims of the paper. Rather, these experiments demonstrate the need for relevant observed variables to enable clustering of representations under hidden confounding. (Relevant variables do not have to cause $Y$ directly but can also help with the clustering because they are correlated with the confounders. Note that $S$ does not causally influence $Y$ in this experiment.)
>
> The main claim of the paper is that boosting can implicitly cluster its representations with respect to a hidden confounder so that the final predictor satisfies the predictive-sufficiency requirement. However, it is crucial that the relevant input covariates be observed for such clustering to be possible.
>
> > The paper acknowledges that the idea of $\alpha$-predictive sufficiency is inspired by by $\alpha$-approximate multi-calibration..
>
> Thank you for the suggestion. We will add a definition of $\alpha$-approximate multicalibration in the revised version of the paper.
>
> We hope our responses have addressed your concerns. We are happy to answer any questions you may have.

---

> > ### Author Response · Authors · 2025-11-27
> > **Request for Feedback on Rebuttal**
> >
> > We would appreciate the reviewer’s feedback on our rebuttal. We believe we have addressed all raised concerns and are happy to answer any further questions.

---

### Official Review · Reviewer_Pav5 · 2025-10-28

**Soundness:** 3
**Presentation:** 3
**Contribution:** 3
**Rating:** 6
**Confidence:** 2

**Summary:**

This paper introduce  $\alpha$-predictive sufficiency and argues that boosting algorithms implicitly identify stable “reference classes” lead to predictors with better OOD behavior under hidden confounding shifts. The authors provide theoretical connections between boosting and predictive sufficiency and validate these claims with synthetic and real-world tabular experiments showing boosting’s robustness in such settings.

**Strengths:**

1. The introduction of $\alpha$-predictive sufficiency is interesting

2. Robust OOD performance under hidden confounding is a pressing problem for real-world tabular data, where many popular OOD methods underperform; the paper directly targets this gap.

**Weaknesses:**

1. The authors focus on tabular datasets. It would be helpful to briefly comment on whether the predictive-sufficiency perspective extends to other modalities like Image

2. Comparisons to other OOD approaches.

3. The author can summarize the contribution of this article in detail and the difference from existing methods to make it easier for readers to understand.

**Questions:**

see weakness

---

> ### Author Response · Authors · 2025-11-21
> **Response to Reviewer Pav5**
>
> Thank you for your insightful review. Below we address each of your comments.
>
> > The authors focus on tabular datasets. It would be helpful to briefly comment on whether the predictive-sufficiency perspective extends to other modalities like Image
>
> Yes, the idea of predictive sufficiency extends to other modalities like images. According to Proposition 4.1, predictive sufficiency can be viewed as a combination of concept shift, residual and a lower-bound on conditional informativeness. Achieving predictive sufficiency is equivalent to simultaneously minimizing concept shift, residual informativeness, and maximizing conditional informativeness. Each of these terms can be optimized separately to achieve predictive sufficiency. For instance, invariance learning methods can be viewed as minimizers of concept shift. Providing environment specific informative covariates can maximize conditional informativeness. Residuals, i.e. $I(\phi(\mathbf{X});Y\mid \hat{Y})$, can be minimized by
> enforcing the independence $Y \perp \phi(\mathbf{X}) \mid \hat{Y}$. By jointly optimizing these terms, it is possible to achieve predictive sufficiency in vision based tasks.
>
> > Comparisons to other OOD approaches.
>
> We perform experiments to compare XGBoost and CatBoost with invariant risk minimization (IRM) [1] and Group DRO [2], two of the popular OOD approaches. We consider the same setup as the synthetic experiment 2 presented in the main paper, where the causal graph is: $U_1\to X, U_1\to Y, X\to Y, U_2\to S, U_2\to Y$, $U_1\sim \mathcal{N}(\mu_e, \sigma_e), U_2\sim \mathcal{N}(\mu_f,\sigma_f)$. Here, $S$ does not cause $Y$ but shares a hidden confounder with $Y$. When only $X$ is used as input, we observe that XGBoost and CatBoost outperform IRM and GroupDRO, indicating that boosting methods are effective under a hidden confounding shift when only partial information about hidden confounding shift is available. When we use both $X,S$ as inputs, only GroupDRO performs on par with boosting, while IRM still performs worse. This also shows that invariance learning is insufficient for generalization under a hidden confounding shift. The results are shown in Figure B6 of the Appendix in the revised paper.
>
> References:
>
> [1] Arjovsky, Martin, et al. "Invariant risk minimization." arXiv preprint arXiv:1907.02893 (2019).
>
> [2] Sagawa, Shiori, et al. "Distributionally robust neural networks for group shifts: On the importance of regularization for worst-case generalization." arXiv preprint arXiv:1911.08731 (2019).
>
> > The author can summarize the contribution of this article in detail and the difference from existing methods to make it easier for readers to understand.
>
> Our main contribution is to theoretically explain the empirical finding that boosting algorithms are surprisingly OOD robust (in particular under prevalent confounding shift).
> To do this, we develop an information-theoretic framework for standard boosting algorithms. We then define $\alpha$-predictive sufficiency, a novel notion that is stronger than multicalibration, and show that for sufficiently small $\alpha$, this condition implies OOD generalization under hidden confounding shift. Next, we connect the information-theoretic view of boosting to $\alpha$-predictive sufficiency.
> Prior work explains boosting’s strengths, particularly on tabular datasets, with variance reduction, handling missing variables, feature selection, and yielding multi-calibrated predictors. Our contribution extends this line of explanation by showing that, from an information-theoretic perspective, boosting returns predictors that are predictive sufficient and thus highly robust to confounding shift.
> We also verify our insights empirically, as we demonstrate that boosting’s representations (obtained by the leaf embeddings) are associated with hidden-confounder information.
>
> We hope our responses have addressed your concerns. We are happy to answer any questions you may have.

---

> > ### Comment · Reviewer_Pav5 · 2025-11-27
> >
> > Thanks for your response. I have decided to keep my score.

---

### Official Review · Reviewer_ZKii · 2025-11-01

**Soundness:** 3
**Presentation:** 3
**Contribution:** 3
**Rating:** 6
**Confidence:** 3

**Summary:**

The paper asks why boosting methods often outperform specialized OOD techniques on real‑world tabular data, especially under hidden confounding shift. It reframes OOD as a reference‑class inference problem and argues that boosting succeeds because it implicitly infers “environments” aligned with latent confounder values. It introduces an information‑theoretic target, α‑predictive sufficiency, measuring whether the residual depends on the (latent) environment.

**Strengths:**

Authors recast OOD under hidden confounding as reference‑class inference and formalizes when a predictor “uses the right class” via α‑predictive sufficiency (Def. 4.2). This helps bridge invariance, multicalibration, and MI‑based viewpoints.
The analysis offers a compelling narrative: boosting’s reweighting plus diverse weak learners implicitly induce environment‑like partitions aligned with confounder strata—consistent with clustering observed in leaf embeddings.
Across synthetic setups and two datasets, increased predictive information and lower sufficiency residuals correlate with better OOD metrics (Table 1), strengthening the paper’s thesis even if not conclusive.

**Weaknesses:**

1. Alg. 1 chooses weak learners by maximizing (I(Y;h(X))) under reweighting and assumes an oracle. CatBoost/XGBoost instead optimize gradient‑based surrogates on losses; no result shows that those procedures approximate the MI‑oracle selection or that their reweighting schedules satisfy Assumption 5.1. The main theorem therefore does not directly cover the widely used algorithms studied empirically.

2. The definition of predictive sufficiency uses a numeric residual; for categorical (Y), this is ill‑posed without an embedding of labels. The paper empirically studies both regression and classification tasks but does not specify how the residual is defined for the latter (e.g., one‑hot, logits, or 0/1 error event).

**Questions:**

1. Can you formalize conditions under which gradient‑boosted tree splitting criteria (e.g., squared‑error or logloss gain) are monotone surrogates for the MI objective in Alg. 1 step 3? Even a lemma relating gain to a lower bound on (I(Y;h(X))) would narrow the theory–practice gap.

2. How exactly is $Y-\hat{Y}$ instantiated when (Y) is categorical (e.g., indicator of misclassification, one‑hot minus probability vector, or log‑odds residual)? Please specify and justify for the MI/KSG estimators.

---

> ### Author Response · Authors · 2025-11-21
> **Response to Reviewer ZKii**
>
> Thank you for your insightful review. Below we address each of your comments.
>
> > Alg. 1 chooses weak learners by maximizing (I(Y;h(X))) under reweighting and assumes an oracle. CatBoost/XGBoost instead optimize gradient‑based surrogates on losses; no result shows that those procedures approximate the MI‑oracle selection or that their reweighting schedules satisfy Assumption 5.1. The main theorem therefore does not directly cover the widely used algorithms studied empirically.
>
> > Can you formalize conditions under which gradient‑boosted tree splitting criteria (e.g., squared‑error or logloss gain) are monotone surrogates for the MI objective in Alg. 1 step 3? Even a lemma relating gain to a lower bound on (I(Y;h(X))) would narrow the theory–practice gap.
>
> After each step of the gradient boosting algorithm, the prediction error (log loss or mean squared error) reduces and moves towards zero. Any boosting algorithm (including CatBoost/XGBoost) should achieve this in order to generalize well. In this context, we rely on the idea that at each step, the addition of a new weak learner increases the overall mutual information between ground truth label and prediction. **Thus, we seek to reformulate classical boosting from being error correction models to information accumulation models.**
>
> Both for regression and classification, the training losses used by modern boosting algorithms measure how uncertain we still are about the label given the current model. In regression, this uncertainty shows up as large residual errors; in classification it appears as high cross-entropy. Whenever the model becomes more accurate, this uncertainty goes down. Reducing uncertainty is exactly the same qualitative goal as increasing mutual information — the less “leftover uncertainty” we have, the more information the prediction carries about the true label. While boosting does not explicitly optimize mutual information, every step it takes is aligned with reducing uncertainty about the label, which is the intuitive counterpart of increasing mutual information. This provides a natural explanation for why CatBoost/XGBoost behave similarly to our MI-oracle and why they achieve strong performance in the presence of hidden confounding shifts.
>
> > The definition of predictive sufficiency uses a numeric residual; for categorical (Y), this is ill‑posed without an embedding of labels. The paper empirically studies both regression and classification tasks but does not specify how the residual is defined for the latter (e.g., one‑hot, logits, or 0/1 error event).
>
> > How exactly is $Y-\hat{Y}$ instantiated when (Y) is categorical (e.g., indicator of misclassification, one‑hot minus probability vector, or log‑odds residual)? Please specify and justify for the MI/KSG estimators.
>
> The quantity $Y-\hat{Y}$ is the difference between the ground truth label and the prediction. We also call it the prediction error. For regression, we can model $Y$ and $\hat{Y}$ to be ground truth and predicted real numbers. For binary classification,  we can model $Y$ and $\hat{Y}$ to be the ground truth label and predicted probability respectively. For theoretical analysis, for a fixed value of $\hat{Y}$, the map $Y \rightarrow (Y-\hat{Y})$ is a bijection. This implies $I(Y-\hat{Y};E\mid \hat{Y}) = I(Y;E\mid \hat{Y})$. We use this formulation in proving Proposition 4.1 that connects $\alpha$-predictive sufficiency and predictive information. For experimental results, instead of evaluating the term $Y-\hat{Y}$ explicitly, we rely on the implementation of  XGBoost and CatBoost models that use gradient based pseudo residual fitting.
>
> We hope our responses have addressed your concerns. We are happy to answer any questions you may have.

---

> > ### Comment · Reviewer_ZKii · 2025-11-26
> >
> > I appreciate the authors' comprehensive response. I have also checked the other points raised by the other reviewers and have decided to maintain my score.

---

### Official Review · Reviewer_91R9 · 2025-11-02

**Soundness:** 2
**Presentation:** 2
**Contribution:** 2
**Rating:** 4
**Confidence:** 4

**Summary:**

The paper introduces an information-theoretic framework linking boosting-type algorithms to out-of-distribution generalization. It defines the notion of $\alpha$-predictive sufficiency, based on conditional mutual information, and presents an iterative “information-boosting” procedure that aims to build predictors satisfying this condition. Theoretical guarantees and empirical results on synthetic and real datasets illustrate the proposed connection between boosting and predictive sufficiency.

**Strengths:**

- The paper proposes an original theoretical connection between boosting and out-of-distribution generalization through an information-theoretic framework. In particular, the introduction of the α-predictive sufficiency concept provides a novel way to formalize robustness to hidden confounding shifts.

- Empirical studies on both synthetic and real datasets offer some qualitative support to the proposed theory.

- The paper is clearly written, with good motivation and intuitive explanations connecting information theory, causality, and ensemble learning. Sometimes the mathematical notation is ambiguous tough.

**Weaknesses:**

- The use of the term *boosting* is largely metaphorical: Algorithm 1 assumes access to an oracle that exactly maximizes a mutual-information objective, rather than invoking a weak learner trained under a reweighted data distribution. As a result, the procedure cannot be interpreted as a genuine boosting algorithm or as a weak-to-strong reduction.

- The main theoretical result (Theorem~5.1) is deterministic and depends on the assumption that the learner performs an exact maximization step. This removes the probabilistic weak-learning premise that underlies classical boosting theory.

- The notation and treatment of the environment variable $E$ are inconsistent: sometimes $E$ is treated as a random variable, elsewhere as a subset or conditioning event, making several definitions (e.g., Def. 5.1) mathematically ambiguous.

- The quantity $I(Y-\hat{Y};E\mid\hat{Y})$ is introduced without a rigorous definition of the residual term $Y-\hat{Y}$, which limits the formal interpretability of the proposed $\alpha$-predictive sufficiency condition.

**Questions:**

Refer to the weaknesses section.

---

> ### Author Response · Authors · 2025-11-21
> **Response to Reviewer 91R9**
>
> Thank you for your insightful review. Below we address each of your comments.
>
> > The use of the term boosting is largely metaphorical: Algorithm 1....
>
> Thanks for the opportunity to clarify this important point. The oracle does not maximize mutual information in an unconstrained manner. Instead, it returns a weak learner $h$ that achieves the highest mutual information within a given hypothesis family $\mathcal{H}$. As can be seen from Line 3 of our algorithm, the maximizer of mutual information is still chosen from the hypothesis space $\mathcal{H}$ of possible weak learners. In traditional boosting, $\mathcal{H}$ is usually a set of linear models or small decision trees. We also seek to maximize the mutual information within this $\mathcal{H}$ of possible weak learners. The only difference is that, instead of measuring the performance of a weak learner by accuracy or squared error, we measure the performance of a weak learner using mutual information.
>
>
> > The main theoretical result (Theorem~5.1) is deterministic and depends on...
>
> Theorem 5.1 is intended to establish convergence of Algorithm 1 to an $\alpha$-predictive sufficient predictor. Definition 5.1 retains the traditional weak-learning premise that weak learners must outperform a baseline model by a small margin in mutual information. Our work reformulates classical boosting theory: We interpret weak learners as information accumulation models rather than error correction models.
>
> As explained in the first paragraph of Section 5 (Lines 307-309), there exist different notions of weak learning. One way of defining a weak learner is as follows: a weak learner $\mathcal{A}$ performs slightly better than a random prediction (1/2) with high probability $1-\delta$:
>
> $(1) \ \mathbb{P}\big(\mathrm{err}_{\mathcal{D}}(\mathcal{A}(S)) \le \frac{1}{2} - \gamma\big) \ge 1-\delta$
>
> However, other notions of weak learning do exist. Our notion of weak learning is based on recent work on boosting, multicalibration, and OOD generalization [1,2]. Specifically, in any restriction of the data distribution $\mathcal{D}$, a weak learner must perform slightly better than a baseline predictor whenever the Bayes predictor performs slightly better than the baseline predictor. To connect our information theoretic notion of weak learning with the above classical notion $(1)$, the assumption of the existence of a weak learner corresponds to $\delta=0$. Also, if the baseline predictor is assumed to provide random predictions (predicting $\frac{1}{2}$), it has a small mutual information with the true labels. Now, the assumption of higher mutual information (compared to the baseline predictor) by a margin $\gamma$ implies that weak learner's error is less than the error made by a baseline predictor. We believe this explanation clarifies how our information-theoretic notion of weak learning is connected to the classical notion in (1).
>
>
> References:
> [1] Globus-Harris, Ira, et al. "Multicalibration as Boosting for Regression." International Conference on Machine Learning. PMLR, 2023.
>
> [2] Wu, Jiayun, et al. "Bridging multicalibration and out-of-distribution generalization beyond covariate shift." Advances in Neural Information Processing Systems. 2024.
>
>
> > The notation and treatment of the environment variable $E$ are inconsistent: sometimes $E$ is treated as a random variable, elsewhere as a subset or conditioning event, making several definitions (e.g., Def. 5.1) mathematically ambiguous.
>
> We agree that there is an ambiguity on how we have used the notation $E$. In the revised manuscript, we use $E$ for environment variable and $\mathcal{E}$ for a set of environments and $e$ for any one specific environment.
>
> > The quantity $I(Y-\hat{Y};E\mid \hat{Y})$ is introduced without a rigorous definition of the residual term $Y-\hat{Y}$, which limits the formal interpretability of the proposed $\alpha$-predictive sufficiency condition.
>
> The quantity $Y-\hat{Y}$ is the difference between the ground truth label and the prediction. We also call it the prediction error. For regression, we can model $Y$ and $\hat{Y}$ to be ground truth and predicted real numbers. For binary classification,  we can model $Y$ and $\hat{Y}$ to be the ground truth label and predicted probability, respectively. We have updated the Definition 4.2 and the paragraph that follows to reflect these points. In our theoretical analysis, for a fixed value of $\hat{Y}$, the map $Y \rightarrow (Y-\hat{Y})$ is a bijection. This implies $I(Y-\hat{Y};E\mid \hat{Y}) = I(Y;E\mid \hat{Y})$. We use this formulation in proving Proposition 4.1 that connects $\alpha$-predictive sufficiency and predictive information. For experimental results, instead of evaluating the term $Y-\hat{Y}$ explicitly, we rely on the implementation of  XGBoost and CatBoost models that use gradient based pseudo residual fitting.
>
> We hope our responses have addressed your concerns. We are happy to answer any questions you may have.

---

> > ### Author Response · Authors · 2025-11-27
> > **Request for Feedback on Rebuttal**
> >
> > We would appreciate the reviewer’s feedback on our rebuttal. We believe we have addressed all raised concerns and are happy to answer any further questions.

---

> > > ### Comment · Reviewer_91R9 · 2025-11-27
> > >
> > > I thank the authors for the detailed rebuttal.
> > >
> > > My concerns still remains.
> > >
> > > I don't understand the notion of weak learner. In classical binary classification, a $\gamma$-weak learner $\mathcal{A}$ for $\gamma \in (0,1/2)$ for $\mathcal{H}$, satisfies (for every distribution $\mathcal{D}$ realized by $\mathcal{H}$)
> > >
> > > $\mathbb{P}(\text{err}_{\mathcal{D}}(\mathcal{A}(S)) \leq 1/2-\gamma) \geq 1-\delta$
> > >
> > > provided that $S$ is at least $n(\delta)$ (this is the sample complexity of the weak learner).
> > >
> > > Could the authors clarify, how their notion of weak learner, generalizes/relate to the classical one?
> > >
> > > Furthermore, I'm not entirely sure I understand the author argument about the lack of probabilistic statements in their aspect.
> > >
> > > Weak learner are usually introduced as their have humble computational requirements (i.e., not solving exact problems and only featuring weak accuracy guarantees), and this gives boosting the computational advantage over ERM and similar approaches. Could the authors elaborate more on this?

---

> > > > ### Author Response · Authors · 2025-11-27
> > > >
> > > > Thank you very much for the response.
> > > >
> > > > > I don't understand the notion of $\gamma$-weak learner. In classical binary classification, a $\gamma$-weak learner $\mathcal{A}$ for $\gamma \in (0,1/2)$ for $\mathcal{H}$, satisfies (for every distribution $\mathcal{D}$ realized by $\mathcal{H}$) $\mathbb{P}(err_{\mathcal{D}}(\mathcal{A}(S)) \leq 1/2 - \gamma) \geq 1-\delta$ provided that $S$ is at least $n(\delta)$ (this is the sample complexity of the weak learner). Could the authors clarify, how their notion of weak learner, generalizes/relate to the classical one?
> > > >
> > > > As explained in the first paragraph of Section 5 (Lines 307-309), there exist different notions of weak learning. As the reviewer pointed out, in binary classification, one way of defining a weak learner is as follows: a weak learner $\mathcal{A}$ performs slightly better than a random prediction (1/2) with high probability $1-\delta$:
> > > >
> > > > $(1) \ \mathbb{P}\big(\mathrm{err}_{\mathcal{D}}(\mathcal{A}(S)) \le \frac{1}{2} - \gamma\big) \ge 1-\delta$
> > > >
> > > > However, other notions of weak learning do exist. Our notion of weak learning is based on recent work on boosting, multicalibration, and OOD generalization [1,2]. Specifically, in any restriction of the data distribution $\mathcal{D}$, a weak learner must perform slightly better than a baseline predictor whenever the Bayes predictor performs slightly better than the baseline predictor. To connect our information theoretic notion of weak learning with the above classical notion $(1)$, the assumption of the existence of a weak learner corresponds to $\delta=0$. Also, if the baseline predictor is assumed to provide random predictions (predicting $\frac{1}{2}$), it has a small mutual information with the true labels. Now, the assumption of higher mutual information (compared to the baseline predictor) by a margin $\gamma$ implies that weak learner's error is less than the error made by a baseline predictor. We believe this explanation clarifies how our information-theoretic notion of weak learning is connected to the classical notion in (1).
> > > >
> > > >
> > > > References:
> > > >
> > > > [1] Globus-Harris, Ira, et al. "Multicalibration as Boosting for Regression." International Conference on Machine Learning. PMLR, 2023.
> > > >
> > > > [2] Wu, Jiayun, et al. "Bridging multicalibration and out-of-distribution generalization beyond covariate shift." Advances in Neural Information Processing Systems. 2024.
> > > >
> > > > > Furthermore, I'm not entirely sure I understand the author argument about the lack of probabilistic statements in their aspect. Weak learner are usually introduced as their have humble computational requirements (i.e., not solving exact problems and only featuring weak accuracy guarantees), and this gives boosting the computational advantage over ERM and similar approaches. Could the authors elaborate more on this?
> > > >
> > > > Thanks for the clarification. As can be seen from Line 3 of our algorithm, we do not solve the exact problem. The maximizer of mutual information is still chosen from the hypothesis space $\mathcal{H}$ of possible weak learners. In traditional boosting, $\mathcal{H}$ is usually a set of linear models or small decision trees. We also seek to maximize the mutual information within this $\mathcal{H}$ of possible weak learners. The only difference is that, instead of measuring the performance of a weak learner by accuracy or squared error, we measure the performance using mutual information.
> > > >
> > > > We hope these responses have addressed your concerns. We would be happy to answer any remaining questions.

---

### Author Response · Authors · 2025-11-21
**Common Response to All Reviewers**

We thank all reviewers for their thoughtful reviews. We are pleased that they found the proposed $\alpha$-predictive sufficiency to be a novel contribution for studying OOD generalization under hidden confounding shift (91R9, ZKii, PaV5, 75bl), that the experimental results support the theory (91R9, ZKii), and that the paper is well written (91R9, 75bl). We address each reviewer’s comments individually below.

---

### Author Response · Authors · 2025-12-03
**Summary Comment to Area Chair**

Dear Area Chair,

We thank you for your time and effort in reviewing our paper and the rebuttal. Below, we provide a summary of the paper, the reviews and our rebuttal.

**Paper summary:** The main goal of our paper is to explain why boosting excels in out-of-distribution (OOD) generalization. To this end, we (i) introduce a novel notion of predictive sufficience, i.e. $\alpha$-predictive sufficiency, that is a stronger notion than multicalibration. (ii) This enables us to propose an information-theoretic framing of boosting together with a corresponding definition of information-theoretic weak learning.
Utilizing these technical innovations, we show that predictive sufficiency implies OOD generalization under hidden-confounding shift, which has been identified as prevalent source of distribution shift in a wide set of benchmarks. Furthermore, boosting achieves predictive sufficiency and hence OOD generalization under hidden confounding shift.
Our experiments corroborate our theory: Predictive sufficiency correlates with performance. Experiments with XGBoost and CatBoost show that clusters in model representations align with hidden confounding variables and this clustering behaviour provides empirical evidence that boosting increases the conditional informativeness required for OOD generalization under hidden-confounding shift.

**Summary of reviewes:** Reviewers acknowledged the importance of the research problem and the novelty of our contributions. In particular, they (i) found $\alpha$-predictive sufficiency to be a novel and useful concept for studying OOD generalization under hidden-confounding shift (91R9, ZKii, PaV5, 75bl), (ii) noted that the experimental results support the theory (91R9, ZKii), and (iii) commented that the paper is well written (91R9, 75bl). Most reviewer questions concerned (i) the connection between information-theoretic boosting and classical boosting, (ii) clarification of residual computation in different settings, (iii) clarification of our theoretical statements, and (iv) editorial improvements,.

**Summary of rebuttal:** We have addressed these points in detail in the rebuttal and we have updated the manuscript to incorporate the reviewers’ suggestions, adding theoretical clarifications, additional experiments, and ablations. We believe these revisions address the reviewers’ concerns and significantly improve clarity and accessibility of our paper.

Thank you again for your time and effort.

---

### Meta-Review · Area_Chair_ERRV · 2026-01-06

**Summary:**

This paper's primary strength is its novel theoretical framework that explains boosting's success in out-of-distribution (OOD) generalization by introducing the concept of α-predictive sufficiency. Their work significantly pushes forward our understanding of OOD generalization and is likely to raise interesting discussions within the community.

The main point of contention was by 91R9 who remained unconvinced by the notion of a weak learner used in this work. However, such determinstic alternative definitions indeed has a long history (e.g. gradient boosting) and I think the skepticism is unwarranted.

Having said this, there are some improvements I recommend the authors to make in their final version -
- Insufficient comparision with classical boosting notions (91R9). While I agree with the authors that there have indeed been alternative frameworks for boosting (e.g. gradient boosting), the authors need to do a better job conveying its evolution and history rather than simply referencing a relatively new paper that is certainly not the first to go beyond the classical weak learner definition.
- Insufficient comparision to other OOD approaches (PaV5) -  while the rebuttal goes some way towards addressing this, very closely related directions such as shortcut learning or simplicity bias are still missing.
- Gap between theory (MI oracle) and practice (gradient boosting using surrogates like log-loss)  (ZKii) needs to be adressed better. It seems that the MI based definition is key to the results here. Do any results transfer for gradient boosting on loss instead?

**Reviewer Concerns:**

- Reviewer 91R9
  - Definition of weak learner - the reviewer and the author went back and forth on the right definition of a weak learner in boosting. Despite the authors citing recent works that use similar non-traditional definitions, the reviewer was unconvinced.
  - Theorem 5.1 lacks probabilistic notions of classical boosting theory - same as above.
  - Definition of residual error. Addressed.


- Reviewer ZKii
  - Gap between theory (MI oracle) and practice (gradient boosting using surrogates like log-loss) - partially addressed. Response gives some intuition why this may be ok, but no theoretical justification.
  - Definition of residual - response clarifies misunderstanding. Addressed.
  - this reviewer explicitly acknowledge the responses.

- Reviewer PaV5
  - Extension from tabular data to images - response argues yes but no actual experiments performed on images.
  - Lack of comparisons to other OOD approaches - authors added IRM and Group DRO baselines. they also provide an overview of comparision to other OOD approaches.


- Reviewer 75bl
  - Real-world data experiments seem synthetic due to artificial shift simulation. While response clarifies why they needed to adjust the test-train split, it does not address the concern that this is still a simulated experimental setup. not addressed.
  - Equation 1 derivation relies on non-peer-reviewed work and unstated assumptions - response clarifies this claim and proof stands on its own. addressed.
  - insufficient background on $\alpha$-approximate multicalibration - authors promised to add this.
  - Numerous other clarifications seeked and addressed.

**Reviewer Scores:**

- 91R9 would likely have retained **4** since they remained unconvinced on their main concern about the definition of the weak learner used here.
- ZKii mentions they retain their **6**
- PaV5 also mentions they retain their **6**
- 75bl had all of their concerns other than realistic experiments addressed and would have increased their score to **6**.

This would result in an average of **5.5**.

---

### Decision · Program_Chairs · 2026-01-26

Accept (Poster)